# Reward history guides focal attention in whisker somatosensory cortex

Deepa L. Ramamurthy [1], Lucia Rodriguez[1,2], Celine Cen[1], Siqian Li[1], Andrew Chen[1] & Daniel E. Feldman [1]✉

Prior reward is a potent cue for attentional capture, but the underlying neurobiology is largely unknown. In a whisker touch detection task, we show that mice flexibly shift attention between specific whiskers on a trial-by-trial timescale, guided by the recent history of stimulus-reward association. Two-photon calcium imaging and spike recordings reveal a robust neurobiological correlate of attention in the somatosensory cortex, boosting sensory responses to the attended whisker in L2/3 and L5, but not L4. Attentional boosting in L2/3 pyramidal cells is topographically precise and whisker-specific, and shifts receptive fields toward the attended whisker. L2/3 vasoactive intestinal peptide (VIP) interneurons are broadly activated by whisker stimuli, motion, and arousal but do not carry a whisker-specific attentional signal, and thus do not mediate spatially focused tactile attention. These findings provide an experimental model of focal attention in the mouse whisker tactile system, showing that the history of recent past stimuli and rewards dynamically engage local modulation in cortical sensory maps to guide flexible shifts in ongoing behavior.

Humans and other animals engage attention to prioritize processing of behaviorally relevant stimuli in complex environments, including for vision[1], audition[2] and touch[3]. Past experiences across different time-scales play an important role in guiding attention[4]. In humans, prior stimulus-reward association is a highly robust cue for attentional capture[5-17], such that perceptual detection or discrimination is selectively enhanced for previously rewarded stimuli, even when those stimuli are no longer important to current goals[14]. This effect is not driven by the physical salience of stimuli (classically termed bottom-up attention) but by their recent association with reward. The effects of reward history on allocation of attentional priority have been termed experience-driven attention[15], value-driven attention[12-14], memory-guided attention[16], or attentional bias by previous reward history[17].

The neurobiology of attention has been studied extensively in non-human primates[1,18], and has been shown to involve boosting of signal-to-noise ratio for neural encoding of attended sensory features across the cortical sensory hierarchy, including primary sensory cortex. However, the fine-scale organization of attentional boosting in

sensory cortex and the neural circuits that control it remain unclear. Mice provide powerful cell-type-specific tools to identify precise neural circuit mechanisms underlying attentional processing. Here, we developed an experimental model of focal attention based on reward history in the mouse whisker tactile system and used it to investigate the neurobiological basis of attention in the sensory cortex.

We studied attentional behavior in head-fixed mice during a Go/NoGo whisker touch detection task, which included Go stimuli on many different whiskers. When the whisker location of a touch stimulus was unpredictable from trial to trial, mice naturally used the history of stimulus-reward association as a cue to guide attention to specific, recently rewarded whiskers. This task generates a rich set of trial histories to probe which stimulus and reward contingencies guide attention, and allows us to track the shifting locus of attention on a trial-by-trial time scale. Using this platform, we identified a robust, spatially focused neural correlate of attention in whisker somatosensory cortex (S1) and tested a major circuit model of attention involving VIP interneurons in sensory cortex. More broadly, our study shows that

[1]Department of Neuroscience and Helen Wills Neuroscience Institute, UC Berkeley, Berkeley, CA, USA. [2]Neuroscience PhD Program, UC Berkeley, Berkeley, CA, USA. ✉e-mail: dfeldman@berkeley.edu

in addition to the well-known role of learned stimulus-reward associations in slowly driving cortical map plasticity, the recent history of stimuli and rewards acts on a fast time scale to dynamically and locally modify cortical sensory maps with corresponding stimulus-specific shifts in perceptual sensitivity.

## Results

To study attention, we developed a head-fixed whisker detection task. Mice had nine whiskers inserted in a piezo array, and on each trial, one randomly selected whisker (Go trials) or no whisker (NoGo trials) was deflected in a brief train. Mice were rewarded for licking during a response window on Go trials (Hits) but not on NoGo trials (False Alarms) (Fig. 1A, B). The task was performed in darkness and incorporated a delay period of 0, 0.5, or 1 s in different mice to separate whisker stimuli from response licks. Go and NoGo trials were randomly intermixed with a variable inter-trial interval (ITI), and whisker identity on Go trials was randomly selected. Thus, mice could not anticipate the upcoming whisker or precise trial timing. Sequential trials could be one Go and one NoGo, two NoGo trials, two Go trials on different whiskers, or two Go trials on the same whisker (Fig. 1C).

Expert mice effectively distinguished Go from NoGo trials, quantified by $d'$ from signal detection theory (Supplementary Fig. S1A). We analyzed behavior in each session during a continuous task-engaged phase that excluded early and late low-performance epochs ($d' < 0.5$) that reflect motivational effects[19]. In expert mice, overall mean $d'$ was $0.923 \pm 0.051$ ($n = 476$ sessions, 22 mice), but local fluctuations in $d'$ regularly occurred over the time course of several trials, suggesting that the recent history of stimuli or rewards may dynamically alter

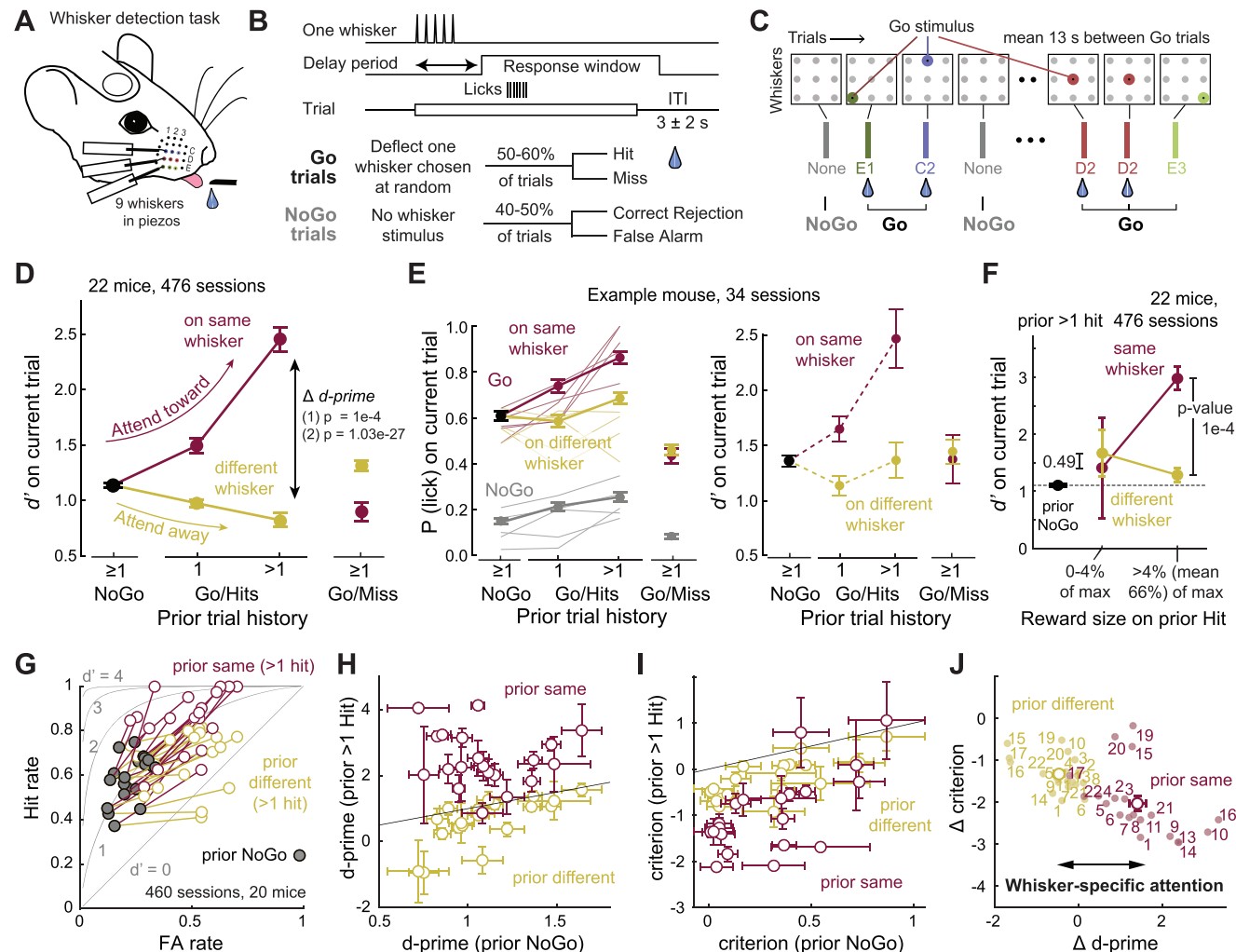

**Fig. 1 | Recent reward history cues spatially specific attention for whisker touch. A** Whisker detection task in head-fixed mice. **B** Trial structure. The delay period was 0, 500, or 1000 ms in different mice (see Supplementary Fig. S1). ITI, Intertrial interval. Bottom, Trial types and outcomes. **C** Example trial sequence with interleaved Go and NoGo trials; whisker identity on Go trials was chosen randomly. A water drop indicates a reward on Hit trials. **D** Mean effect of trial history on detection sensitivity (d-prime; 476 sessions, 22 mice). Prior Hits to the same whisker (maroon) increased $d'$ for detecting that whisker on the current trial ('attend toward'), while prior Hits to a different whisker (yellow) decreased $d'$ ('attend away'). *P*-values: (1) Permutation tests (prior same 1 Hit vs prior NoGo: $p = 1e-4$, prior same >1 Hit vs prior NoGo: $p = 1e-4$, prior different >1 Hit vs prior NoGo: $p = 1e-4$, prior same >1 Hit vs prior different >1 Hit: $p = 1e-4$), (2) Linear mixed effects model, prior history class: $p = 1.03e-27$; sex: $p = 0.155$). **E** Same effect in a single example mouse. Left, underlying effect on hit rate for 5 example sessions (thin lines) and all 34 sessions (thick lines). Right, $d'$ for this mouse from 34 sessions. **F** $d'$ did not increase after unrewarded hits or very small rewards to the same whisker (< 4%: $p = 0.49$, > 4%: $p = 1e-4$); *p*-values for difference in d' between prior same vs. different conditions (permutation test). **G** Effect of > 1 prior hits on current trial Hit Rate and False Alarm (FA) rate. Connected symbols show data from individual mice (460 sessions, 20 mice). Gray lines, receiver-operating characteristic (ROC) curves for different $d'$ levels. Prior same whisker hits drive a whisker-specific increase in hit rate over FA rate and improve $d'$. **H, I** Behavioral shifts in $d'$ (**H**) and criterion *c* (**I**) across individual mice; criterion changes were less whisker-specific. **J** Changes in $d'$ ($\Delta d'$) and *c* ($\Delta c$) per mouse included in **G**–**I** (number, mouse identity). We focus on $\Delta d'$ to index whisker-specific attention. See also Supplementary Fig. S1.

sensory detection behavior (Supplementary Fig. S1A). To examine this, we classified each current Go trial based on the history of whisker stimuli and reward on the immediately preceding trials. We used trial history categories of: (i) prior NoGo, (ii) prior Hit to the same whisker as the current trial, (iii) prior Hit to a different whisker, (iv) prior Miss to the same whisker, and (v) prior Miss to a different whisker. Current NoGo trials were classified into categories of prior NoGo, prior Hit (to any whisker), or prior Miss (to any whisker). We separately tracked trials preceded by a single Hit trial from those preceded by multiple sequential prior Hits ("prior >1 Hit"; Supplementary Fig. S1B). For each trial history category, we quantified detection sensitivity (d′) and criterion (c) from the mouse's hit rate and false alarm rate for all current Go or NoGo trials with that trial history in the session.

## Reward history cues focal attention in the whisker system

Prior trial history strongly influenced detection on the current trial. When the prior trial was a NoGo, mice detected the current Go whisker with $d' = 1.13 \pm 0.02$ (termed $d'_{NoGo}$, mean ± SEM across 476 sessions, 22 mice), which we consider baseline detection sensitivity. $d'$ for detecting a given Go whisker was elevated following 1 prior Hit to the same whisker, and even more so following multiple consecutive Hits to the same whisker ($d'_{>1HitSame} = 2.45 \pm 0.11$, $p = 1.0e\text{-}4$ vs $d'_{NoGo}$, permutation test). In contrast, $d'$ was reduced following one or multiple prior Hits to a different whisker than the current trial ($d'_{>1HitDiff} = 0.82 \pm 0.06$, $p = 1.0e\text{-}4$ vs $d'_{NoGo}$, permutation test) (Fig. 1D, E). $d'$ after multiple prior Hits to the same whisker ($d'_{>1HitSame}$) was substantially greater than after multiple prior Hits to a different whisker ($d'_{>1HitDiff}$) (Fig. 1D). Thus, recent Hits engage a whisker-specific boost in detection, evident as a whisker-specific increase in $d'$, termed $\Delta d'$ (Fig. 1D). This effect was found across mice with 0, 0.5, or 1-sec delay period ($p = 1.0e\text{-}4$ for $d'_{>1HitSame}$ vs $d'_{>1HitDiff}$ in each case), so these data were combined for behavioral analyses (Supplementary Fig. S1D).

Trial history-dependent boosting of detection required the conjunction of prior stimulus plus reward, because $d'$ did not increase after a prior Miss to the same whisker ($d'_{MissSame} = 0.90 \pm 0.09$, $p = 4.2e\text{-}3$ relative to $d'_{NoGo}$). Boosting also failed to occur if the mouse licked to the prior Go but received no reward (i.e., unrewarded hits) or a very small reward (< 4% of maximal reward volume ($d'_{>1HitSame}$ vs $d'_{>1HitDiff}$, $p = 0.49$, permutation test) (Fig. 1F). Thus, boosting was not driven by prior whisker deflection alone, or stimulus-evoked licking, but by whisker-reward association on recent trials.

We interpret this effect as whisker-specific attention, because it shares characteristic features of stimulus-specific attention documented in primates during detection tasks[20–24]. This involves both increased sensitivity (d′) and a shift in decision criterion (c, also referred to as response bias), with the whisker-specific $\Delta d'$ reflecting increased Hit rate following >1 prior Hit to the same whisker relative to >1 prior Hit to a different whisker (p = 1.0e-4, permutation test; Fig. 1G). The whisker-specific increase in $d'$ was observed across mice (Fig. 1H; $p = 1.0e\text{-}4$, permutation test). We also observed overall shifts towards more liberal criterion values[20,22,23] ($\Delta c$) due to the increased False Alarm rate (Fig. 1I). $\Delta c$ had a modest whisker-specific component ($c_{>1HitSame}$ vs $c_{>1HitDiff}$, $p = 0.002$, permutation test) due to the influence of Hit rate on criterion calculation. On average, the whisker-specific shift in sensitivity was larger than the whisker-specific shift in criterion (Fig. 1J; $\Delta d'_{>1HitSame} = 1.4 \pm 0.20$, $\Delta d'_{>1HitDiff} = -0.47 \pm 0.11$, $p = 1.0e\text{-}4$, permutation test; $\Delta c_{>1HitSame} = -2.03 \pm 0.18$, $\Delta c_{>1HitDiff} = -1.34 \pm 0.08$, $p = 7.0e\text{-}4$, permutation test).

Mice trained on all 3 delay periods showed the whisker-specific boost in $d'$ (Supplementary Fig. S1C, D). Reaction times on Go trials were reduced by prior Hit trials, as expected for attention (assessed in mice without a delay period), but this effect was not whisker-specific, which may reflect a ceiling effect (Supplementary Fig. S1E). The task interleaved whisker deflections of varying amplitude, and whisker-specific boosting of $d'$ was greatest for current Go trials with low-amplitude (weak) deflections, as expected for attention (Supplementary Fig. S1F). Trial history effects were consistently observed across different variations of the task (used in different mice) in which we manipulated the stimulus probability of each whisker in blocks, or manipulated the probability of sequential same-whisker Go trials to make prior same Hit histories more likely (Supplementary Fig. S1G, H). Trial history effects were driven by stimulus-reward association, not by stimulus salience or reward probability, because whisker deflections were physically identical and reward probability was always 100% for each whisker.

Behavioral shifts in $d'$ exhibited the hallmark effects of focused attention: spatial specificity, temporal specificity, and flexible targeting. Spatially, $d'$ was boosted most strongly by prior Hits to the same whisker, more weakly by prior Hits to an immediate same-row or same-arc neighbor, and not at all by prior Hits to a diagonally adjacent or more distant whisker (Fig. 2A). Thus, boosting is somatotopically organized. Temporally, attentional boosting fell off with the time interval between Go trials (which varied due to variable ITI and

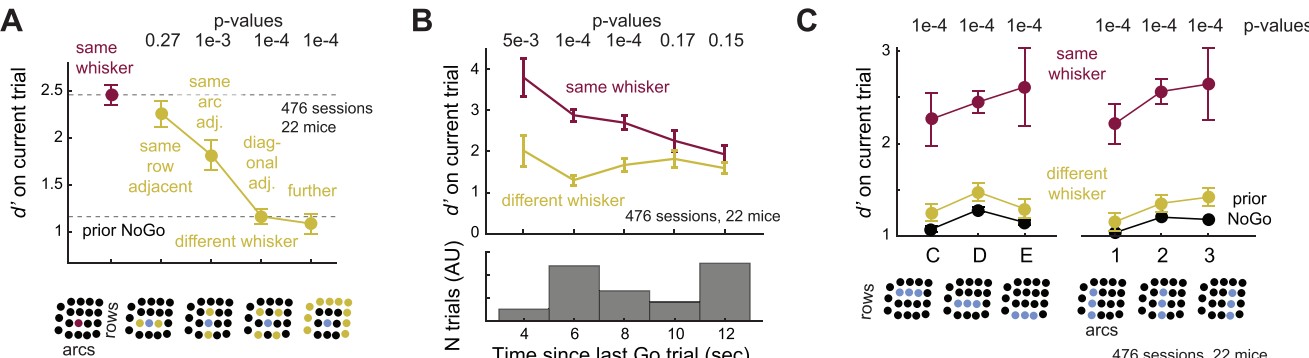

**Fig. 2 | Spatiotemporal characteristics of attention cued by reward history.** **A** Spatial gradient of the attention effect. The facial position of the prior Hit whisker is plotted on the x-axis, relative to the current trial whisker. P-values are for difference from prior same (same row: p = 0.27, same arc: 1e-3, diagonal: 1e-4, further: 1e-4, permutation test). Bottom, each position shown schematically. **B** Temporal profile of the whisker-specific attention effect (3–5 s: 5e-3, 5–7 s: 1e-4, 7–9 s: 1e-4, 9–11 s: 0.17, > 11 s: 0.15, permutation test). *P*-values are from permutation tests (FDR-corrected). Bottom, Inter-Go-trial intervals sampled in the task. **C** Flexible targeting of attention. x-axis represents identity of whisker in current Go trial, grouped into C, D, or E-row whiskers (left), or arcs 1, 2 or 3 (right). For all of these whiskers, prior reward history boosts detection in a whisker-specific way. Bottom, each row or arc shown schematically. *P*-values are for prior same vs prior different >1 hit (p = 1e-4 for all whisker positions, permutation test). Data are presented as mean ± SEM across sessions in all panels, unless otherwise specified. Conventions as in Fig. 1. Data from 476 sessions, 22 mice in all panels. See also Supplementary Fig. S1.

intervening NoGo trials), and subsided after ~10 s (Fig. 2B). Enhancement of *d'* was flexibly shifted to different whiskers in an interleaved manner, and had similar magnitude when cued by trial history to any of the 9 whiskers in rows B-D or arcs 1–3 (Fig. 2C). Thus, mice use recent history of stimulus-reward association to dynamically boost sensory detection of spatially specific whiskers on a rapid trial-by-trial time-scale, consistent with attentional enhancement[1,18,20–23,25,26]. These properties strongly resemble attentional capture guided by reward history in humans and non-human primates[4–17,24].

### Whisker-specific attention is not mediated by whisker or body movement

To test whether trial history effects involve whisker or body movement, we extracted these movements, plus pupillary dilations related to arousal[27,28], from behavioral videos of 9 mice (74 sessions) using DeepLabCut[29] (Supplementary Fig. S2A). Reward retrieval at the end of Hit trials was associated with whisker movement, body movement (detected from platform motion), and pupil dilation that slowly subsided during the ITI before the next trial (Fig. 3A). The magnitude of movement and pupil dilation during the ITI was greater after 1 or >1 prior Hits, relative to prior NoGo, but was identical for prior same and prior different conditions (Fig. 3A and Supplementary Fig. S2B). Thus, mice exhibited increased motion and arousal following prior Hits. During the subsequent trial, whisker stimulation evoked modest whisker and body motion during the stimulus period, and these were also heightened after prior Hits ($p = 1e{-}4$, permutation test), indicating that behavioral arousal and motion effects from prior Hits persisted into subsequent trials, but did not differ between prior same whisker and prior different conditions ($p = 0.34$ and $0.76$, permutation test Fig. 3A, B).

To test whether active whisker movement contributed to the whisker-specific $\Delta$ *d'* effect, we paralyzed whisker movements with Botulinum toxin B (Botox) injection in the vibrissal pad in 4 task-expert mice. Paralysis was maintained over 7-8 days, and history effects were compared between standard sessions prior to Botox and the Botox sessions. The average whisker-specific shift in behavioral *d'* and *c* following >1 prior hit did not differ between Botox and non-Botox sessions (Fig. 3C, D), demonstrating that whisker-specific attentional effects do not require active whisker movement. Thus, although prior hits also engage increases in whisker motion, body motion and arousal, whisker-specific attentional effects were independent of these effects on global behavior state.

### Neural correlates of focal attention in L2/3 PYR cells in S1

The somatotopic precision of attentional effects on behavior (Fig. 2A) suggests a neural basis in a somatotopically organized brain area like S1. We performed 2-photon imaging in S1, using Drd3-Cre;Ai162D mice that transgenically express GCaMP6s in L2/3 PYR cells. Mice performed the task with a delay period that separated whisker-evoked responses (analyzed 0–799 ms after stimulus onset) from later licks and rewards, and any trials with early licks were excluded. Imaging fields were localized in the S1 whisker map by post-hoc cytochrome oxidase staining for whisker barrel boundaries in L4 (Fig. 4A, B). 61% of L2/3 PYR neurons were whisker responsive, and trial history modulated whisker responses for many individual neurons. For example, in Fig. 4C, neurons increased their response to the C3 whisker after >1 prior hit to that same whisker, but not after >1 prior hit to a different whisker.

On average, L2/3 PYR cells responded to a given Go whisker when the prior trial was a NoGo, responded more strongly following 1 prior Hit to the same whisker ($p = 1e{-}4$), and even more after >1 prior Hit to the same whisker ($p = 1e{-}4$). This boosting of sensory responses did not occur when the prior trial was a Miss to the same whisker, or >1 Hit to a different whisker (Fig. 4D, E, $n = 6$ mice, 70 sessions, 6906 PYR cells, 4–19 sessions per mouse, $118.7 \pm 5.96$ cells (range: 3–213) per session).

Thus, recent stimulus-reward association modulated whisker-evoked $\Delta$F/F in L2/3 PYR cells in a way that closely resembled the behavioral attention effect (Fig. 4E vs Fig. 1D). On current NoGo trials, no whisker stimulus was presented, and $\Delta$F/F traces were largely flat, except for NoGo trials following prior Hit trials, which exhibited a surprising rising $\Delta$F/F signal. We interpret this as an expectation or arousal effect, which parallels the increased FA rate on these trials (Fig. 1G). These effects were evident in single example fields (Supplementary Fig. S3A), and persisted for the duration of the whisker response, even after the lick-free period (Supplementary Fig. S3B). History-dependent boosting of whisker responses was most evident in whisker-responsive cells (defined from trials after a prior NoGo), and did not occur in cells that were non-responsive after a prior NoGo (Supplementary Fig. S3C, D). Boosting was evident when analysis was restricted to current Hit trials, confirming that it represents sensory modulation and not decision (Supplementary Fig. S3E). Boosting after prior Hits was whisker-specific in all 6/6 mice, but its magnitude varied across mice (Fig. 4F) and correlated with the magnitude of the behavioral attention effect measured by $\Delta$ *d'* in each mouse (Fig. 4G).

We quantified the attention effect in individual cells using three attention modulation indices (AMI). $AMI_{>1HitSame{-}NoGo}$ and $AMI_{>1HitDiff{-}NoGo}$ quantify the change in whisker-evoked response observed after >1 prior Hit to the same (or different) whisker vs after a prior NoGo. Most cells showed positive $AMI_{>1HitSame{-}NoGo}$ values and negative $AMI_{>1HitDiff{-}NoGo}$ values, indicating up- and down-modulation of response magnitude by the identity of the prior whisker Hit. $AMI_{>1HitSame{-}>1HitDiff}$ compares response magnitude after >1 prior hit to the same whisker vs >1 prior hit to a different whisker. This was shifted to positive values for most L2/3 PYR cells, indicating whisker-specific attentional modulation (Fig. 4H, I). This was reproducible across individual mice (Supplementary Fig. S3F). Modulation of whisker-evoked responses for all cells sorted by $AMI_{>1HitSame{-}>1HitDiff}$ is shown in Supplementary Fig. S3G.

History-dependent boosting was somatotopically restricted in S1. After >1 prior hits to a given whisker, responses to an immediate same-row adjacent neighboring whisker were boosted strongly, those to an immediate same-arc neighbor were boosted less, and those to diagonal adjacent neighbors or further whiskers were boosted the least or not at all. This somatotopic profile of $\Delta$F/F boosting strongly resembled the somatotopy of behavioral *d'* boosting (Fig. 4J). Spatially within S1, >1 prior hits to a reference whisker boosted whisker-evoked $\Delta$F/F to that whisker most strongly for L2/3 PYR cells within the reference whisker column and in the near half of the neighboring column, and weakly or not at all beyond that. This somatotopically constrained boosting was not observed for >1 prior hits to a different whisker, or following a miss to the reference whisker (Fig. 4K). Analysis of individual cell AMI confirmed somatotopically precise boosting (Supplementary Fig. S4A). This defines the spatial profile of the trial history-based 'attentional spotlight' in S1 as boosting responses to the cued whisker within a region of 1.5 columns width in L2/3. At the center of the spotlight, the representation of the attended whisker is boosted within its own column (Supplementary Fig. S4B, C).

### Attention involves shifts in receptive fields toward the attended whisker

In primates, attention not only increases sensory response magnitude and signal-to-noise ratio in sensory cortex, but can also shift neural tuning toward attended stimuli[30–32]. We tested whether attention involves receptive field shifts by L2/3 PYR cells in S1. We analyzed all whisker-responsive cells (defined from the Prior NoGo condition) relative to the boundaries of the nearest column. We calculated the mean receptive field across 9 whiskers centered on the columnar whisker (CW) when the prior trial was a NoGo (Fig. 5A, black traces in center panel). We then recalculated the receptive field, for these same cells, for >1 Prior Hit to each of the surround whiskers (purple traces in

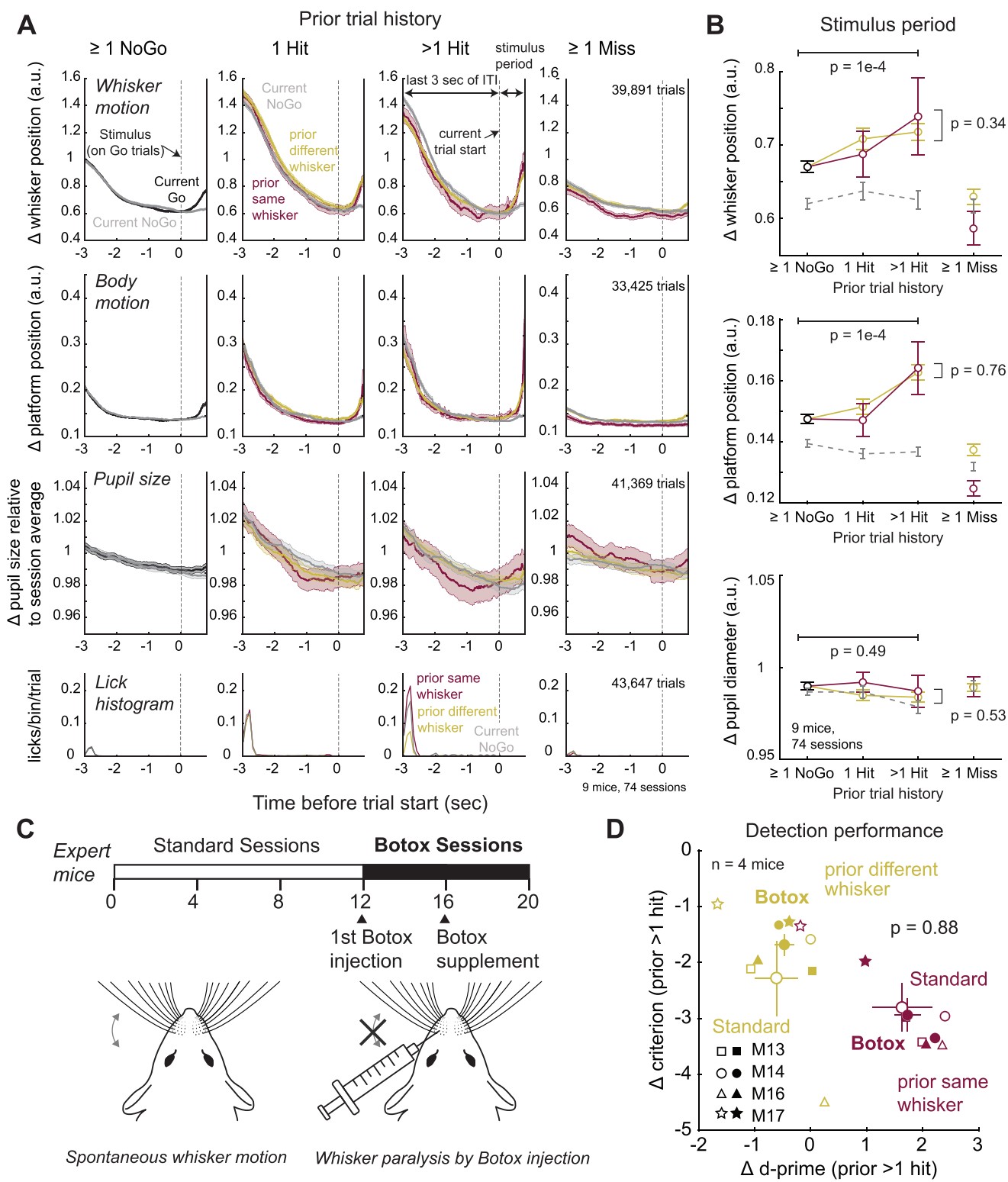

outside panels). For many whiskers, recent stimulus-reward association shifted the mean receptive field toward the attended whisker or nearby whiskers (Fig. 5A). To quantify this effect, we calculated the tuning center of mass (CoM) across the 3 by 3 whisker array for these mean receptive fields. History-based cueing to an attentional target whisker generally shifted tuning CoM towards that whisker, as evident from comparing rostral vs. caudal or upward vs. downward attentional shifts (Fig. 5B). The magnitude of CoM shifts was smaller when attention was cued upward or within the CW row [(r,-) and (c,-) positions].

These differences were not explained by known experimental factors, but could reflect spatial asymmetries in attentional effects[33] on whisker touch (Fig. 5B).

To measure receptive field shifts in individual neurons, we quantified the shift in CoM for each responsive cell along an attentional axis from the CW to the attentional target whisker, which was one of the 8 surrounding whiskers (Supplementary Fig. S5A, B). In the >1 prior Hit condition, the mean CoM shift along the attentional axis was $0.12 \pm 0.04$ ($n = 173$ cells with sufficient whisker sampling, $p = 3.6e\text{-}3$,

**Fig. 3 | Whisker motion, body motion, and arousal do not account for whisker-specific behavioral effects. A** Mean whisker motion, platform motion (proxy for body motion), and pupil size traces, from DeepLabCut analysis across 43,647 trials, 74 sessions in 9 mice. Each panel shows the last 3 s of the intertrial interval (ITI) period after the prior trial, plus the current trial stimulus period. Traces and shading are mean ± SEM across all trials. Dashed line, stimulus onset (Go trials) or dummy piezo onset (NoGo trials). Pupil size was normalized within each session to the mean pupil size across the whole session. Bottom row, lick histogram. **B** Mean whisker movement, body movement, and pupil size change during the stimulus period. Prior Hits increased stimulus-evoked whisker and body motion on subsequent trials (Δ whisker motion, prior >1 hit same vs prior NoGo: $p = 1e\text{-}4$, prior >1 hit same vs >1 hit different: $p = 0.39$; Δ body motion, prior >1 hit same vs prior NoGo: $p = 1e\text{-}4$, prior >1 hit same vs >1 hit different: $p = 0.39$; Δ pupil area, prior >1

hit same vs prior NoGo: $p = 0.49$, prior >1 hit same vs >1 hit different: $p = 0.53$). p-values are for >1 Prior Hit vs Prior NoGo (top), and >1 Prior Hit Same vs >1 Prior Hit Different (right) (permutation test). Error bars show SEM across trials. **C** Design of Botox experiment. Behavior was assayed on an average of 12 sessions pre-Botox injection, and 7 sessions post-Botox whisker paralysis. **D** Reward history-dependent attention effect in each of the 4 mice tested, for standard sessions (before Botox, open symbols) and Botox sessions (filled symbols). M, Mouse numbers as in Fig. 1J. Large points are mean ± SEM across mice. Conventions as in Fig. 1J. Whisker paralysis did not alter the mean whisker-specific d-prime effect or criterion effect ($p = 0.88$, permutation test comparing same vs. different shifts in Δd′ for standard and Botox sessions. Paired differences in same vs. different Δd′ shifts across session types were tested against zero). See also Supplementary Fig. S2.

---

one-sample permutation test vs. mean of 0), indicating a tuning shift toward the attended whisker. A range of tuning shifts were observed, with significantly more cells showing CoM shifts towards the attended whisker than away from it (62% vs. 38%, $p = 2.3e\text{-}3$, binomial exact test for difference from 0.5) (Fig. 5C).

When the mouse attended to the CW, responses to that whisker within its S1 column increased, but there was no obvious shift in tuning peak (Fig. 5D). Instead, neurons increased their responsiveness to the CW, and decreased their responsiveness to the strongest SWs (Fig. 5E), thus increasing their preference (tuning sharpness) for the CW (Fig. 5F). Thus, attentional cueing involves receptive field shifts, tuning width changes, and modulation of whisker response magnitude, as observed during classical attention studies in primates[18,30–32]. The mean tuning changes, combining all three of these effects, that occur in a S1 column as the mouse shifts attention to each nearby whisker are shown as receptive field contour plots in Supplementary Fig. S5C.

### Attention boosts population decoding of recently rewarded whiskers

Is the magnitude of attentional boosting of L2/3 PYR responses sufficient to improve neural coding of attended whisker stimuli on single trials? To test this, we built a neural population decoder that uses logistic regression to predict the presence of any whisker stimulus (i.e., stimulus detection) from single-trial ΔF/F for each whisker-responsive cell in a single imaging field. The weight of each cell in the decoder was fit by ridge regularization (see Methods). Each field spanned ~1–1.5 columns within the 9-whisker region of S1, typically centered on the column representing the center whisker in the piezo array. A separate decoder was trained on each session ($n = 70$ sessions, 6 mice), and performance was assessed from held-out trials (Fig. 6A). Because S1 is somatotopically organized, we observed moderate, above-chance performance for detecting any of the 9 whiskers from mean activity in a single field (relative to NoGo trials or shuffled data), strong performance for detecting the field best whisker (fBW) that is topographically matched to the field location, and weak ability to detect non-topographically aligned whiskers (non-fBWs) (Fig. 6B). Imperfect performance of the decoder reflects the fact that neural population activity (ΔF/F) on NoGo trials had some overlap with Go trials, especially Miss trials. As a result, when decoder performance was analyzed separately for behavioral Hit, Miss, FA and CR trials, stimulus detection (of any whisker) by the decoder was better for current Hit trials than current Miss trials (Supplementary Fig. S6).

We examined decoding as a function of prior trial history. Detection of any whisker stimulus from Go trials was improved following >1 Hit to the same whisker, relative to prior NoGo ($p = 1e\text{-}4$, permutation test) or to >1 prior Hit to a different whisker ($p = 1e\text{-}4$, permutation test) (Fig. 6C). This boost in decoding performance did not occur when the decoder was tested only on fBW trials but did was prominent for non-fBW trials (Fig. 6D, E). Overall, multiple prior hits to the same whisker boosted non-fBW detection, but not fBW detection, and did not boost decoder false alarms on NoGo trials (Fig. 6F).

Thus, attention to a whisker strengthens its encoding on single trials in S1. The lack of improvement in decoding attended fBWs likely reflects the strong coding of these whiskers under baseline conditions, such that attentional boosting of CW responses (Supplementary Fig. S4) does not further improve detection. We interpret the lack of improvement in whisker decoding after a single Hit (Fig. 6C–E) to mean that 1 prior Hit boosts whisker responses only modestly relative to single-trial variability within the imaging field.

### Attentional modulation of neural coding with Neuropixels spike recordings

To examine attentional modulation of S1 neural coding at finer temporal resolution and across layers, we recorded extracellular spiking in S1 using Neuropixels probes[34] that spanned L1-6 (Fig. 7A). Mice performed a modified version of the task in which Go stimuli were distributed over only 4 or 5 whiskers, rather than 9, to enable adequate sampling of each history condition per session. Behaviorally, mice performing the task with 4-5 whiskers showed the whisker-specific Δ $d′$ attention effect, but at a lower magnitude due to the smaller number of whiskers (Supplementary Fig. S7A). All sessions included the CW for the recording site, plus 3 nearby whiskers. We spike sorted to identify single units, classified units as regular-spiking (RS) or fast-spiking (FS), and assigned laminar identity based on CSD analysis of local field potentials (Supplementary Fig. S7B–F). Many single units showed history-dependent modulation of whisker-evoked spiking (Fig. 7B).

S1 units responded to each deflection in the stimulus train. On average for L2/3 RS units (3 mice, 47 cells, 3-4 sessions per mouse, mean 15.7 ± 1.90 [range 11–18] cells per mouse), whisker-evoked spiking was boosted in Go trials after >1 prior Hit to the same whisker, relative to >1 prior Hit to a different whisker (Fig. 7C, D). This effect did not occur after prior Miss trials, and firing on NoGo trials was not significantly regulated (Fig. 7D). L4 RS units (37 cells, mean 12.3 ± 3.93 [range 3 19] cells per mouse) showed only a slight trend for history-dependent modulation that did not reach significance (Fig. 7E, F). L5a/b RS units (112 cells, mean 37.3 ± 8.85 [range 26-59] cells per mouse) showed a whisker-specific attentional boost similar to L2/3 (Fig. 7G, H).

To examine heterogeneity across units, we calculated AMI for each unit. Most L2/3 RS units responded more strongly after prior Hits to the same whisker relative to prior NoGo (as measured by $AMI_{>1HitSame\text{-}NoGo}$) and more weakly after prior Hits to a different whisker relative to prior NoGo (as measured by $AMI_{>1HitDiff\text{-}NoGo}$). This whisker-specific attentional effect, evident as the separation between $AMI_{>1HitSame\text{-}NoGo}$ and $AMI_{>1HitDiff\text{-}NoGo}$ distributions, was not present in L4, and was weaker in L5a/b (Fig. 7I). These laminar trends were also apparent in $AMI_{>1HitSame\text{-}>1HitDiff}$, which was shifted positively in L2/3 relative to L5a/b and L4 units (Fig. 7J). Calculating the mean AMI across neurons confirmed whisker-specific attentional shifts in L2/3 and L5, but not in L4 (Fig. 7K; ($AMI_{>1HitSame\text{-}NoGo}$ vs $AMI_{>1HitDiff\text{-}NoGo}$, L2/3: $p = 0.03$, L4 $p = 0.65$, L5a/b: $p = 0.01$; $AMI_{>1HitSame\text{-}>1HitDiff}$, L2/3: $p = 0.031$, L4 $p = 0.51$, L5a/b: $p = 0.06$, permutation test). This suggests

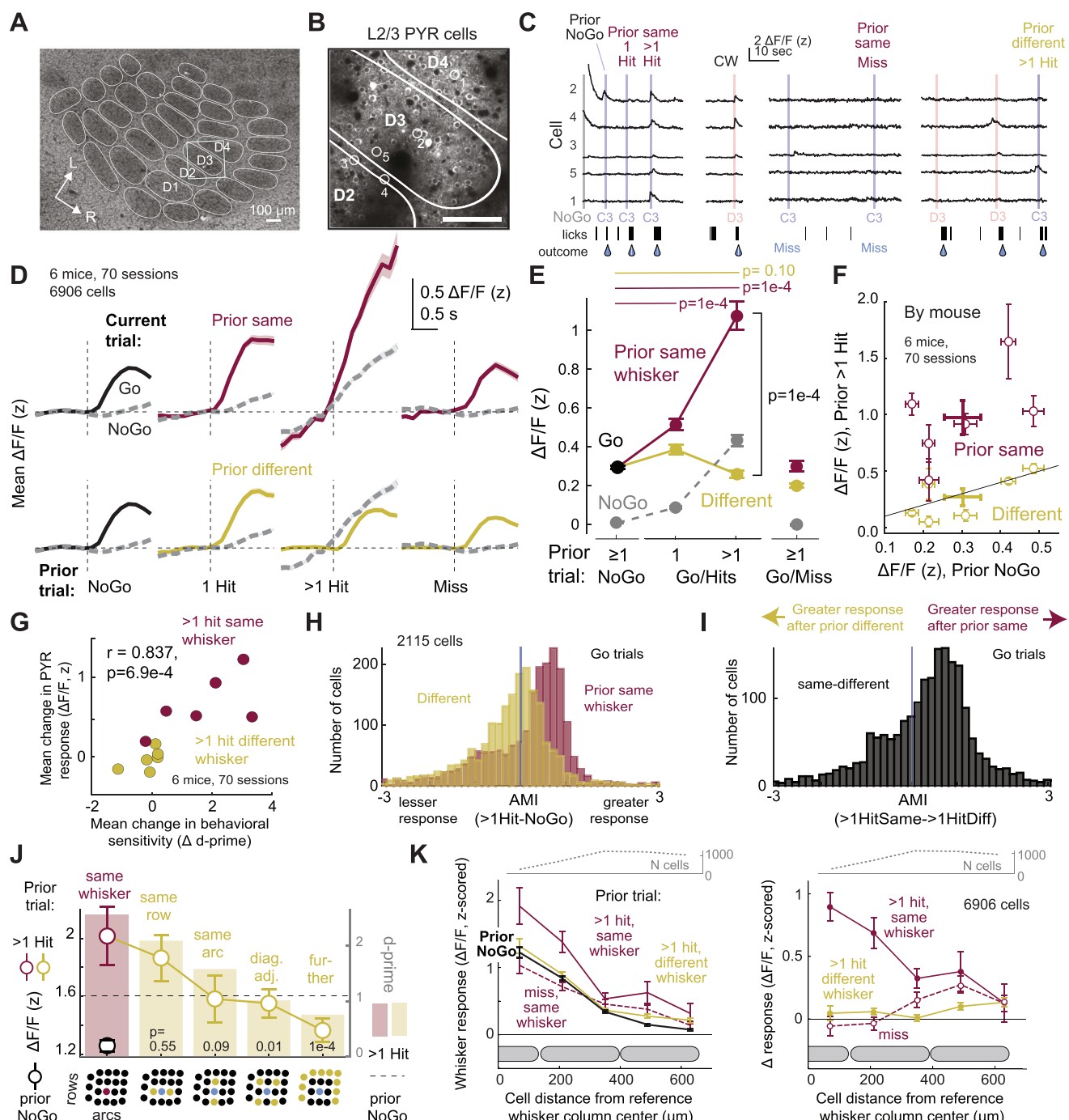

**Fig. 4 | Neural correlates of attentional capture in L2/3 pyramidal cells in S1. A, B** Example imaging field in S1 centered on the D3 column in a Drd3-Cre;Ai162D mouse (1/70 sessions, 6 mice). L2/3 pyramidal (PYR) cells express GCaMP6s. Scale bar = 100 μm. **C** Example trials showing strong responses to columnar whisker D3 but weak responses to surround whisker C3, which were strongly modulated by prior trial history. C3 responses increased following multiple prior hits to C3 (left), but not other history conditions (right). **D** Mean ΔF/F traces by trial history; solid = Go whisker, dashed = NoGo. **E** Quantification of whisker-evoked ΔF/F by trial type (Go/NoGo) and history. *P*-values (permutation test, FDR-corrected): prior same 1 Hit vs prior NoGo: *p* = 1e-4, prior same >1 Hit vs prior NoGo: *p* = 1e-4, prior different >1 Hit vs prior NoGo: *p* = 0.1, prior same >1 Hit vs prior different >1 Hit: *p* = 1e-4. Linear mixed effects model, prior history class: *p* = 1e-4; sex: *p* = 0.878. **F** Mean ΔF/F modulation by mouse. **G** Correlation between history-based modulation of PYR whisker responses and behavioral d-prime across mice (*r* = 0.837,

*p* = 6.9e-4, Pearson's correlation). **H** AMI$_{>1HitSame-NoGo}$ and AMI$_{>1HitDiff-NoGo}$ for each cell. Positive values indicate a greater response than prior NoGo. **I** AMI$_{>1HitSame->1HitDiff}$ for each cell. **J** Boosting of whisker-evoked ΔF/F responses as a function of somatotopic offset between prior and current trial whisker. P-values from permutation tests (FDR-corrected). Bars show behavioral d-prime for the same mice (subset of data from Fig. 2A). **K** Somatotopic organization of attentional capture. Left, ΔF/F evoked by a reference whisker as a function of cell position relative to its column center. When calculated from Prior NoGo trials (thick black trace), this defines the classic point representation of a single whisker. This is boosted in prior same >1 Hit trials, but not prior different >1 Hit or prior same miss trials. Right, same data shown as difference relative to Prior NoGo trials. Data shown as mean ± SEM across cells in all panels. Conventions as in Fig. 1. See also Supplementary Figs. S3 and S4.

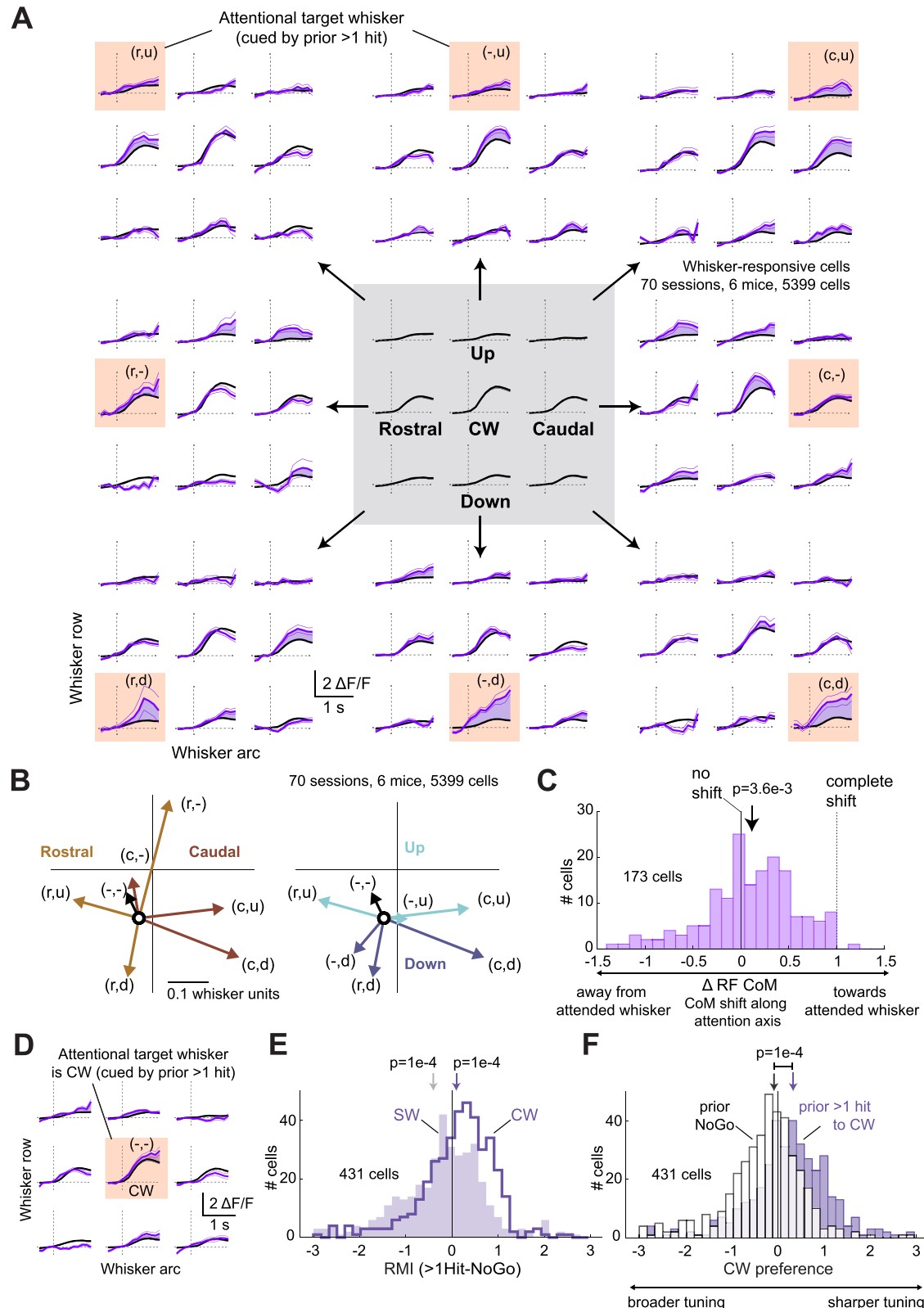

**A** Attentional target whisker (cued by prior >1 hit)

Whisker-responsive cells
70 sessions, 6 mice, 5399 cells

Whisker row

2 ΔF/F
1 s

Whisker arc

**B** 70 sessions, 6 mice, 5399 cells

Rostral    Caudal

(r,-)    (c,-)

(r,u)    (-,-)    (c,u)

(r,d)    (c,d)

0.1 whisker units

Up

(r,u)    (-,-)    (c,u)
(-,u)

(-,d)
(r,d)    (c,d)

Down

**C**

no shift    p=3.6e-3    complete shift

173 cells

# cells

away from attended whisker    Δ RF CoM CoM shift along attention axis    towards attended whisker

**D** Attentional target whisker is CW (cued by prior >1 hit)

Whisker row

(-,-)

CW

2 ΔF/F
1 s

Whisker arc

**E**

p=1e-4    p=1e-4

SW    CW

431 cells

# cells

RMI (>1Hit-NoGo)

**F**

p=1e-4

prior NoGo    prior >1 hit to CW

431 cells

# cells

CW preference

broader tuning    sharper tuning

---

that history-dependent attentional modulation is not simply inherited from the thalamus but has a cortical component.

**VIP interneurons do not carry a simple "attend here" signal**

We used reward history-based attention in S1 to investigate the candidate involvement of VIP interneurons in attentional control. VIP cells are known to disinhibit PYR cells to increase PYR sensory gain during arousal, locomotion, and whisking[35–39]. For attention, this same VIP circuit has been hypothesized to be activated by long-range (e.g., top-down) inputs, and to act to amplify local PYR responses to selected sensory features[38,39] (Fig. 8A). Whether VIP cells mediate goal-directed attention is still unclear[40,41], and their involvement in history-based attention has not been tested. We used 2-photon imaging from L2/3 VIP cells in VIP-Cre;Ai162 mice to ask whether VIP cells are activated

**Fig. 5 | Attentional cueing involves receptive field shifts toward attended whiskers. A** Mean whisker-evoked ΔF/F trace for all whisker-responsive cells in all imaged columns. Center, mean ± SEM (across $N = 5399$ cells) for trials when prior trial was NoGo, separated by the identity of the current trial whisker. This reports the average whisker tuning curve for these neurons, in the absence of attentional cueing. Outer panels, the whisker responses measured when prior trial history was >1Hit to the indicated attentional target whisker (thick purple trace is mean, thin traces show ± SEM). Purple fill is drawn between mean traces to aid visualization. r, u, c, d denote rostral, up, caudal, or down from the columnar whisker (CW). **B** Mean tuning center-of-mass (CoM) when the prior trial was NoGo (black circle) vs after prior >1 Hit to each of the indicated whiskers as defined in panel A. CoM coordinate system is shown in Supplementary Fig. S5A. Vectors are color-coded for whether

the target whisker was rostral, caudal, up, or down from the CW. **C** Magnitude of CoM shift along the attention axis, as defined in Supplementary Fig. S5B. Negative values are shifts away from the attended whisker. The mean CoM shift was significantly greater than zero ($p = 3.6e\text{-}3$, permutation test). **D** Mean whisker receptive field when prior trial was >1 hit to the CW (purple), relative to when prior trial was NoGo (black). Format as in panel (**A**). **E** Response modulation index (RMI) for modulation of CW and SW responses when the attention is directed to the CW. RMI shows that CW responses were increased ($p = 1e\text{-}4$, permutation test), while mean SW responses were decreased ($p = 1e\text{-}4$, permutation test). **F** Attention to the CW increases preference of neurons to the CW relative to the top three SWs, thus narrowing their overall tuning width ($p = 1e\text{-}4$, permutation test). See also Supplementary Fig. S5.

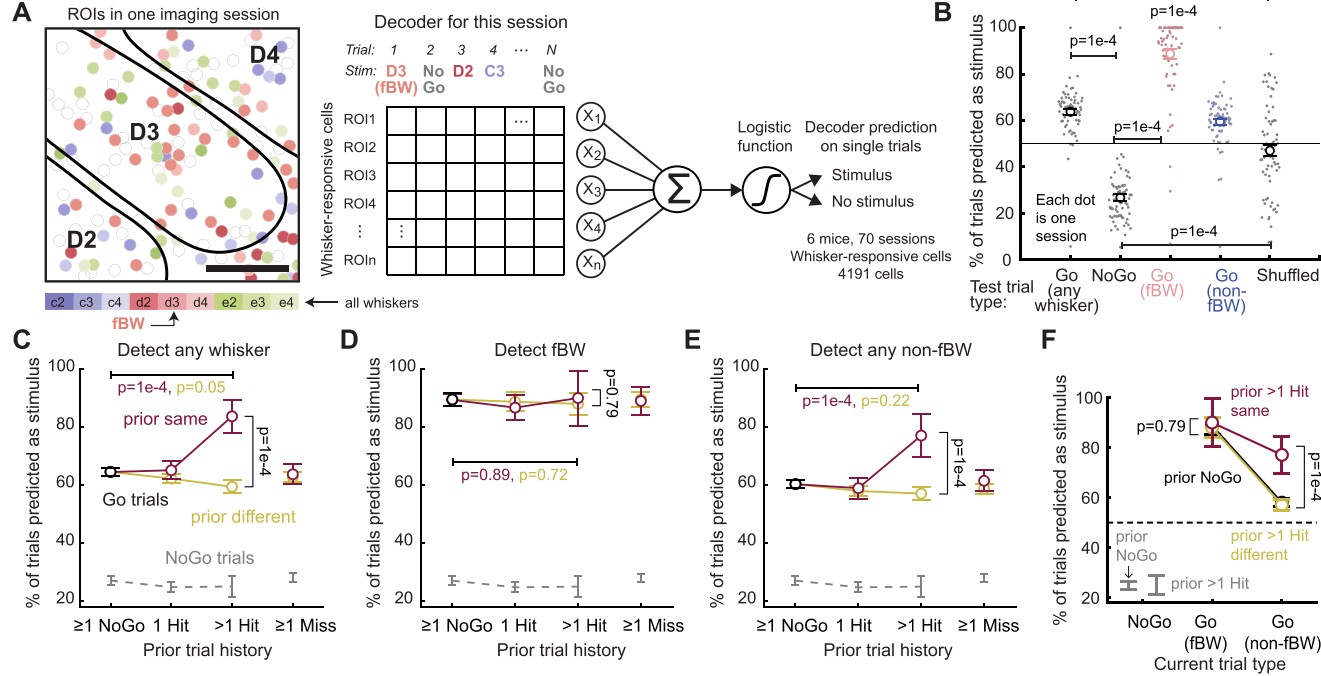

**Fig. 6 | Attentional cueing improves neural decoding of attended whiskers on single trials. A** Neural decoder design. Left, example field showing L2/3 PYR cells tuned to different best whiskers intermixed in each column, consistent with prior studies[84]. Right, weights for individual neurons were fit using ridge regression 10-fold cross-validation was used to fit a logistic regression predicting stimulus presence (any whisker Go trial) or absence (NoGo trial). **B** Decoder performance determined from held-out trials (6 mice, 70 sessions). Each dot is one session. Mean ± SEM across sessions is overlaid on the data. Decoder performance is relatively low for any whisker, because many trials are Go trials for whiskers that are not strongly represented in the imaging field. Decoder performance for field best whisker (fBW) trials is high because this whisker is strongly represented in the imaging field. **C** Mean decoder performance separated by trial history type, when the current trial is any Go whisker (maroon or yellow) or a NoGo trial (gray dash).

Solid lines with circle markers show decoder performance across any whisker on the current trial. (All trials, prior same >1 Hit vs prior NoGo: $p = 1e\text{-}4$, prior different >1 Hit vs prior NoGo: $p = 0.05$, prior same >1 Hit vs prior different >1 Hit: $p = 1e\text{-}4$). **D** Same as (**C**), but for decoder performance when the current trial is an fBW Go trial or a NoGo trial. **E** Same as (**C**), but for decoder performance when the current trial is a non-fBW Go trial or a NoGo trial. **F** Summary of attentional modulation of decoding accuracy for the prior >1 Hit trial history condition. >1 prior Hit to a whisker improves single-trial decoding for non-fBW whiskers, but not for the fBW, in each field (non-fBW: $p = 1e\text{-}4$, fBW: $p = 0.79$). Data are presented as mean ± SEM across sessions in all panels. *P*-values are for prior same vs prior different >1 hit (permutation test). *P*-values from permutation (FDR-corrected), in all panels. See also Supplementary Fig. S6.

---

when mice direct attention to a particular whisker column in S1 (Fig. 8B).

VIP cells in S1 are activated by arousal (indexed by pupil size), whisker and body movement, and goal-directed licking during this whisker detection task[42]. These behaviors all peak at the end of Hit trials, as mice retrieve rewards, and then systematically decline during the ITI, which ends with a 3-sec lick-free period that is required to initiate the next trial. As a result, VIP cell ΔF/F falls systematically during the ITI after Hit trials, correlated with these behavioral variables, and falls less after NoGo or Miss trials (Fig. 8C). This creates a declining baseline for ΔF/F of VIP cells at the beginning of each current trial. VIP cells in S1 also show robust whisker stimulus-evoked ΔF/F

transients, which ride on this declining baseline[42]. To test whether whisker-evoked VIP responses are greater when reward history cues attentional capture, we calculated mean ΔF/F traces ($n = 7$ mice, 103 sessions, 1843 VIP cells, 7–31 sessions per mouse, mean $23.8 \pm 0.61$ [range 9–56] cells per mouse) as a function of trial history. Baseline (prestimulus) ΔF/F declined more steeply on trials following prior Hits than following prior NoGo or Miss, as expected. Superimposed on this, and clearest after detrending the baseline, whisker-evoked ΔF/F was also increased after prior Hit trials. However, this was not whisker-specific (Fig. 8D–F). AMI analysis confirmed increased responsiveness for most VIP cells in both prior >1 Hit same and prior >1 Hit different conditions, but no whisker-specific attention effect (the

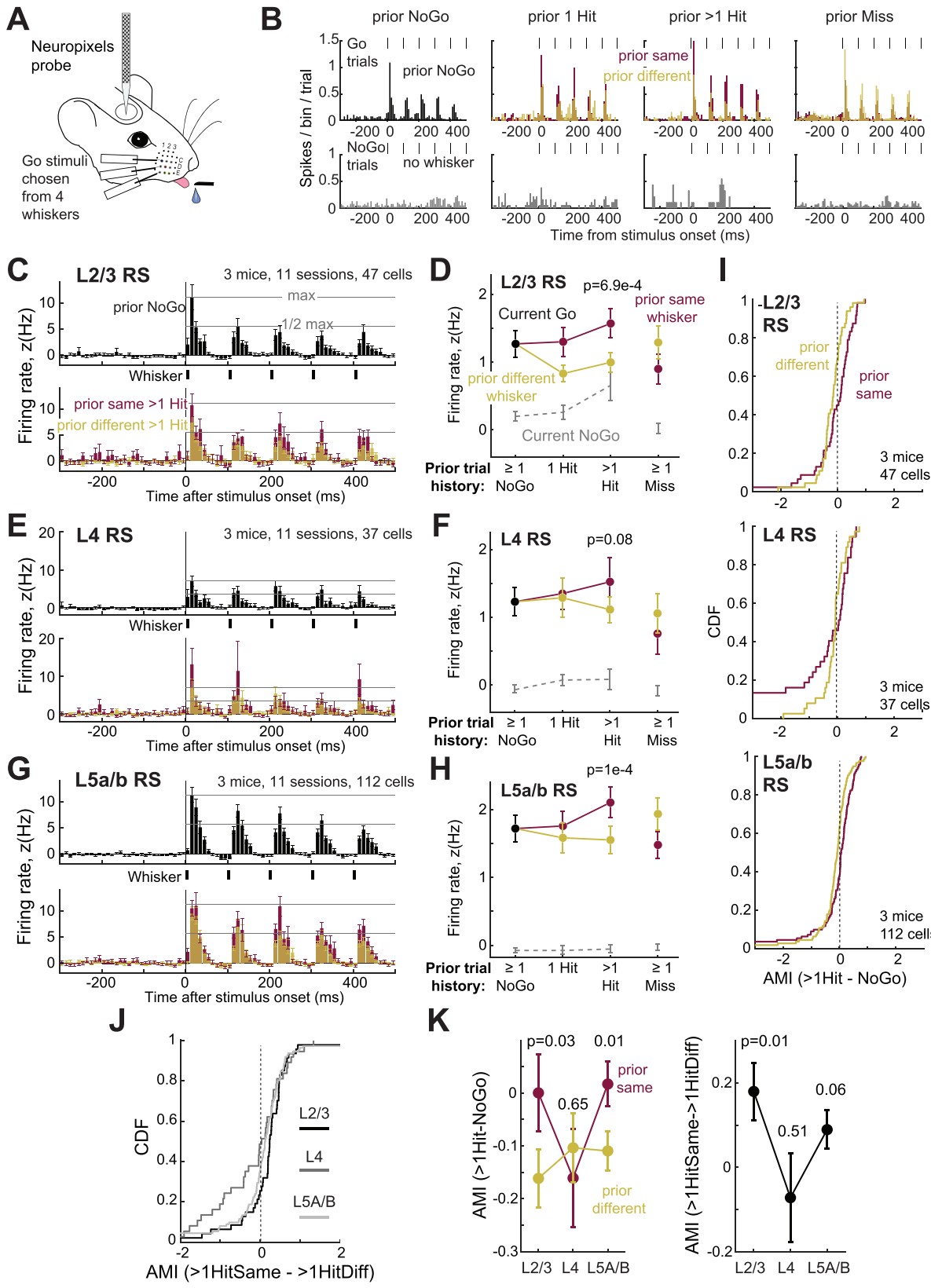

AMI$_{>1HitSame->1HitDiff}$ index was peaked at 0; Fig. 8G, for all cells individually see Supplementary Fig. S8).

Together, these findings indicate that VIP cells as a population do not carry a whisker-specific attention signal, but do exhibit a general increase in activity with multiple prior Hits to any whisker that is consistent with global arousal and motion effects. Although we cannot rule out that a subpopulation of VIP cells may carry a whisker-specific "attend here" signal, the VIP population as a whole does not.

## Discussion

Attention captured by recent history cues[4–17], including stimulus-, reward-, and choice-history, provides a powerful model to study

**Fig. 7 | Attentional effects on extracellular single-unit spiking in S1.**
**A** Neuropixels recording during the whisker detection task. **B** Mean PSTH for an example L2/3 regular spiking (RS) unit recorded in the D1 whisker column, across all Go trials (D1, C2, delta, and gamma whiskers, top) and NoGo trials (bottom). Whisker-evoked responses were boosted when prior trial history was 1 or > 1 Hit to the same whisker (maroon), and reduced when prior trial was > 1 Hit to a different whisker (yellow). **C**, **E**, **G** Layer-specific analysis for RS units of population mean PSTH (10 ms bins) for Go trials, by trial history (3 mice, L2/3: 47 cells, L4: 37 cells, L5: 112 cells). **D**, **F**, **H** For the same units, population average whisker-evoked response

on current Go trials (500-ms window) by trial history, or for current NoGo trials in an equivalent window (prior same > 1 Hit vs prior different > 1 Hit, L2/3: $p$ = 6.9e-4, L4: 0.08, L5a/b: 1e-4, permutation test). **I** Cumulative distribution (CDF) of attention modulation indices (AMI), $AMI_{>1HitSame-NoGo}$ and $AMI_{>1HitDiff-NoGo}$ values for RS units by layer. **J** Cumulative distribution of $AMI_{>1HitSame->1HitDiff}$ by layer. **K** Mean AMI values by layer for the same units ($AMI_{>1HitSame-NoGo}$ vs $AMI_{>1HitDiff-NoGo}$, L2/3: 0.03, L4: 0.65, L5a/b: 0.01; $AMI_{>1HitSame->1HitDiff}$, L2/3: 0.01, L4:0.51, L5a/b: 0.06). Error bars show SEM across cells in all panels. $P$-values from permutation tests (FDR-corrected) in all panels. See also Supplementary Fig. S7.

mechanisms of attention. In our paradigm, enhanced detection (Δ $d'$) of whisker stimuli was driven by recent whisker stimulus-reward association, had the defining features of selective attention (spatially focused, flexibly allocated, and temporally constrained)[4,20–23]. Behavioral and neural effects in our study were whisker-specific, and thus did not correspond to a global arousal or motion effect[43]. They were not explained by priming, which occurs in response to stimulus presentation without reward association, does not require detection of the priming stimulus, and typically has a short (<100 ms) duration[44].

Attentional boosting in our study was not bottom-up attention, because it was not driven by physical stimulus salience (whisker stimuli had the same physical stimulus magnitude), and it was not top-down attention, because it was automatically engaged and not goal-directed (i.e., whisker stimuli were all equally rewarded, so it did not increase overall reward rate in the task). These results align well with automatic attentional capture by reward history in humans[4–17], which is theorized to represent a category of attentional processes often called "selection history"[8–10] distinct from classical top-down and bottom-up attention. Rodents are well known to exhibit history-dependent response biases (i.e., shifts in decision criterion for behavioral choices) in perceptual decision-making tasks[45–49], including serial dependence[45,49,50], contraction bias[45,46,49], adaptation aftereffects[51], win-stay/lose-switch strategies[52–54] and choice alternation[52]. Our findings show that mice also use prior reward history to prioritize sensory processing, through stimulus-specific shifts in perceptual sensitivity ($d'$), in addition to shifts in decision criterion ($c$).

The neural correlates of attention have been primarily studied in non-human primates, and include increased sensory-evoked spike rate[18,55,56], reduced variability[18], neural synchrony modulation[57], and changes in receptive fields[18,30–32] including in receptive size, boosting of peak responses, and receptive field shifts. In primates, these effects are greatest at higher levels of the sensory hierarchy but also occur in primary sensory cortex[18,58]. The precise spatial organization of these coding effects in sensory cortex has been unknown and has important implications for identifying the neural control circuits for attention. On the macroscopic scale, human brain imaging and focal pharmacological inactivation studies in non-human primates indicate that spatial attention in vision is retinotopically organized within visual cortical areas[1,18,59–62], including in a study of value-guided attention that showed enhanced sensory-evoked activation driven by recent prior reward in human V1[63]. But the precise spatial organization of attentional modulation in sensory cortex (i.e., the spatial profile of the spotlight of attention) has not been known. Importantly, our task design (in which we track spontaneous behaviors, and separate stimulus and lick response windows with a delay period) allowed us to distinguish attentional signals from global arousal and motion signals, which dominate neural activity during behavior and are widespread across cortical areas[43].

We took advantage of S1 whisker map topography[64] to quantitatively define the precise spatial structure of the attentional spotlight relative to anatomical cortical columns in S1. Attentional capture boosted sensory responses to the attended whisker in a region comprising that whisker's column plus the near half of the surrounding columns. In this region, whisker-evoked spike rate and receptive field peak increased (in the central attended column), receptive fields

sharpened (also in the attended column), and receptive fields shifted toward the attended whisker (for cells in surrounding columns). Together, this increased total neural activity evoked by the attended whisker, both by increasing the number of PYR cells responding to that whisker and by elevating the number of spikes per cell. This somatotopically restricted boosting[59–65] is distinct from the spatially broad modulation of sensory responses that occurs across entire cortical areas (or multiple areas) in response to global behavioral state (e.g., arousal indexed by pupil size, active whisker movement for S1, or locomotion for V1)[43]. Our results show that attentional boosting can be flexibly targeted with a precision of ~ 300 μm in cortical space for stimulus-specific modulation of the neural code. Thus, neural control circuits for attention (which may involve feedforward, local, feedback, or neuromodulatory circuits) must operate with this spatial precision. In addition to this somatotopically precise boost in PYR cell sensory gain, recent rewards also drove a generalized increase in activity on subsequent NoGo trials, which was not whisker-specific and may contribute to the non-whisker specific shift in behavioral criterion.

History signals in mouse cortex have not previously been described for attention, but have been identified in posterior parietal cortex (PPC) and orbitofrontal cortex (OFC) during decision making[66–68] and in reversal learning[69–72]. S1 receives instructional signals from OFC that are necessary for reversal learning[71], but whether this pathway plays a role in attention cued by recent reward history is unknown.

The neural mechanisms and control circuits for attention remain poorly understood, and likely differ between different forms of attention. We found that attention to touch modulates PYR sensory responses in L2/3 and L5a/b but not L4, suggesting either an intracortical origin or a thalamic origin in secondary thalamic nuclei like the posterior medial nucleus (POm), which projects to L2/3 and L5a. Thalamic control of attention has been implicated in visual and tactile cross-modal attention tasks in mice[73], as well as some non-human primate studies[74]. We tested one major circuit model[37–41] for attentional boosting in sensory cortex, that long-range inputs amplify pyramidal (PYR) cell sensory responses by activating local VIP interneurons in sensory cortex[37,75]. L2/3 VIP interneurons receive local, feedforward, and feedback glutamatergic input, as well as by neuromodulatory input, and inhibit other cortical interneurons to disinhibit PYR cells, thus boosting PYR sensory responses[37,75]. VIP interneurons are activated by global behavioral state (e.g., locomotion and whisking), by spontaneous arousal during quiet wakefulness[27,35,36], and by top-down contextual signals[38]. Top-down input from the anterior cingulate cortex (ACC) to VIP cells in sensory cortex has been suggested to mediate top-down attentional effects on sensory processing[37]. However, recent studies have questioned this model[40,76], finding that L2/3 VIP modulation of PYR activity is orthogonal to attentional effects in a cross-modal attention task[40]. We tested the potential involvement of VIP cells in focal attention by asking whether VIP cell activity is enhanced in attended columns, as required if these cells contribute to boosting of PYR cell responsiveness. We found that L2/3 VIP cells, at least as a full population, do not carry this whisker-specific "attend-here" signal, but instead show general, non-whisker-specific activation in response to any prior Hit, consistent with an arousal- or global behavior-related signal. This suggests VIP cells are more engaged in

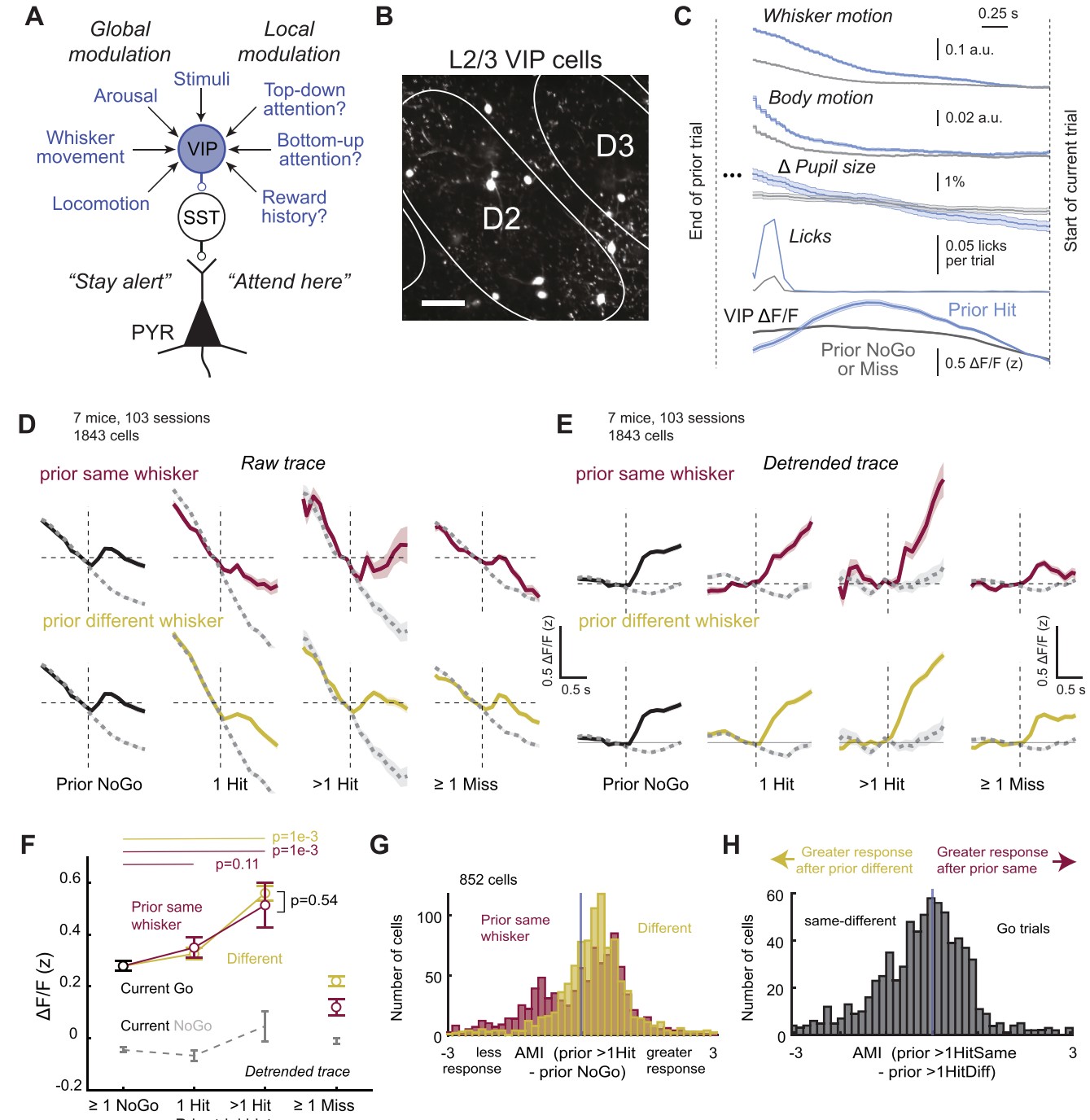

**Fig. 8 | L2/3 VIP cells carry a general arousal signal, but not a whisker-specific attentional signal. A** Circuit model for potential vasoactive intestinal peptide-expressing (VIP) cell role in arousal, movement, and attentional modulation in sensory cortex. SST – somatostatin-expressing interneuron. **B** Example L2/3 imaging field in a VIP-Cre;Ai162D mouse (1/103 sessions, 7 mice). Scale bar = 50 μm. **C** Mean ΔF/F traces during the last 3 s of the intertrial interval (ITI) after the prior trial, for prior rewarded (prior 1 Hit or prior >1 Hit trials) and prior unrewarded trials (prior NoGo or prior Miss). Traces are zeroed to the last 2 frames before stimulus onset. Dynamics of licking behavior, pupil size, whisker position, and body position in the ITI period. The declining baseline, evident on both prior rewarded (blue) and unrewarded (gray) trials, is due to VIP cell encoding of arousal, whisker motion, and body motion which all decline in the inter-trial interval as described in a previous study[42], and shown in Fig. 3A. Prior rewarded trials show a steeper peak

and steeper decline due to licking and reward consumption at the end of the prior trial. **D** Mean ΔF/F traces by trial history; solid = Go whisker, dashed = NoGo. Conventions as in Fig. 4D. **E** Same data as in (**D**) after detrending (see "Methods"). Whisker-evoked responses are non-selectively modulated by prior hits on either the same or different whiskers, consistent with a global arousal effect but not whisker-specific attention. **F** Quantification of whisker-evoked ΔF/F by trial type (Go/NoGo) and history (prior same 1 Hit vs prior NoGo: $p = 0.11$, prior same >1 Hit vs prior NoGo: $p = 1e-3$, prior different >1 Hit vs prior NoGo: $p = 1e-3$, prior same >1 Hit vs prior different >1 Hit: $p = 0.54$, permutation test. Linear mixed effects model, prior history class: $p = 0.507$; sex: $p = 0.753$). Conventions as in Fig. 4E. **G** AMI$_{>1HitSame-NoGo}$ and AMI$_{>1HitDiff-NoGo}$ for each cell. Positive values indicate greater response than prior NoGo. **H** AMI$_{>1HitSame->1HitDiff}$ for each cell. Data shown as mean ± SEM across cells in all panels. See also Supplementary Fig. S8.

global modulation of whisker sensory responsiveness during arousal and motion, and not whisker-specific attention cued by reward history.

Attention to whisker touch cued by recent reward history in mice provides a useful paradigm for studying the neurobiological mechanisms of focal attention. This model complements recent visual tasks that aim to study top-down[77–80] and bottom-up[81–83] attention in head-fixed mice. Together, these paradigms can reveal the extent to which common vs. distinct neurobiological mechanisms are engaged in different forms of attention, and across different sensory modalities.

## Methods

### Animals

All methods followed NIH guidelines and were approved by the UC Berkeley Animal Care and Use Committee. The study used 22 adult mice. These included 7 Drd3-Cre;Ai162D mice and 10 VIP-Cre;Ai162D mice (used for behavior and 2-photon imaging), and 5 offspring from Drd3-Cre x Ai162D crosses (genotype not determined) used in extracellular recording experiments. VIP-Cre (JAX # 10908) and Ai162D mice (JAX # 031562) were from The Jackson Laboratory. Drd3-Cre mice were from Gensat MMRRC (strain number 034610).

Mice were kept in a reverse 12:12 light cycle and were housed with littermates before surgery and individually after cranial window surgery (ambient temperature: 20–26°, humidity: 30-70%). Mice were roughly evenly divided between male and female, and no sex differences were found for the results reported here. Behavioral, imaging, and analysis methods were as described in Ramamurthy et al., 2023[42], and are here described more briefly. Results reported for 7 VIP-Cre;Ai162D mice and 3 Drd3-Cre;Ai162D mice are new analyses which include data from the dataset reported in Ramamurthy et al., 2023.

### Surgery for behavioral training and 2-photon imaging

Mice (2-3 months old for all strains) were anesthetized with isoflurane (1–3%) and maintained at 37 °C. Dexamethasone (2 mg/kg) was given to minimize inflammation, meloxicam (5–10 mg/kg) for analgesia, and enrofloxacin (10 mg/kg) to prevent infection. Using sterile technique, a lightweight (< 3 g) metal head plate containing a 6 mm aperture was affixed to the skull using cyanoacrylate glue and Metabond (C&B Metabond, Parkell). The headplate allowed both head fixation and 2-photon imaging through the aperture. Intrinsic signal optical imaging (ISOI) was used to localize either C-row (C1, C2, C3) or D-row (D1, D2, D3) barrel columns in S1[84], and a 3 mm craniotomy was made within the aperture using a biopsy punch over either the C2 or D2 column. The craniotomy was covered with a 3 mm diameter glass coverslip (#1 thickness, CS-3 R, Warner Instruments) over the dura and sealed with Metabond to form a chronic cranial window. Mice were monitored on a heating pad until sternal recumbency was restored, given subcutaneous buprenorphine (0.05 mg/kg) to relieve postoperative pain and then returned to their cages. After the mice recovered for a week, behavioral training began.

### Behavioral task

To motivate behavioral training, each mouse received 0.8–1.5 mL of water daily, calibrated to maintain 85% of pre-training body weight. Mice were weighed and observed daily. Behavioral training sessions took place 5–7 days per week. For behavioral training, the mouse was head-fixed and rested on a spring-mounted stage[42,85]. Nine whiskers were inserted in a 3 × 3 piezo array, typically centered on a D-row or C-row whisker. Piezo tips were located ~ 5 mm from the face, and each whisker was held in place by a small amount of rubber cement. A tenth piezo was present near the 3 × 3 array but did not hold any whisker ("dummy piezo"). A capacitive lick sensor (for imaging experiments) or an infrared (IR) lick sensor (for extracellular recording experiments) detected licks, and water reward (mean 4 µl) was delivered via a solenoid valve. Mice were transiently anesthetized with isoflurane (0.5–2.0%) at the start of each session to enable head-fixation and

whisker insertion, after which isoflurane was discontinued, and behavioral testing began after the effects of anesthesia had fully recovered. Behavior was performed in the dark with 850 nm IR illumination for video monitoring. Masking noise was presented from nearby speakers to mask the piezo actuator sounds. Task control, user input and task monitoring were performed using custom Igor Pro (WaveMetrics) routines and an Arduino Mega 2560 microcontroller board.

**Training stages.** A series of training stages (1–5 days each) were used to shape behavior on the Go/NoGo detection task. In Stage 1, mice were habituated to the experimental rig and to handling. In Stage 2, mice were head-fixed and conditioned to lick for water reward at the port. In Stage 3, mice received a reward (cued by a blue light) for suppressing licks for at least ~ 3 s, termed the Interlick Interval (ILI) threshold. Stage 4 introduced whisker stimulation for the first time, with 50% Go trials (whisker stimulation) and 50% NoGo trials (no whisker stimulus), with automatic reward delivery in the response window on Go trials (i.e., classical conditioning). The dummy piezo was actuated on all NoGo trials, so that any unmasked piezo sounds did not provide cues for task performance. The Go whisker randomly chosen from among 9 possible whiskers. This stage ended when mice shifted licks in time to occur before reward delivery. In Stage 5, training switched to operant conditioning mode, and mice were required to lick in the response window (0–300 ms after whisker stimulus onset) to receive a reward. There was no delay period at this stage. Learning progress was tracked by divergence of Go/NoGo lick probability. In Stage 6, the delay period was introduced. To do this, we introduced a trial abort window in which licking during the stimulus presentation period caused a trial to be canceled without reward, to discourage licking during the stimulus presentation. Simultaneously, we implemented a ramp-plateau reward gradient within the response window, so that later licks resulted in a larger reward. Over the course of this stage, the trial abort window was gradually lengthened, and the time of reward plateau was gradually increased. Learning progress was tracked by the gradual increase in median trial first lick time. Stage 7 represented the final whisker detection task, which included completely randomized Go/NoGo trials, a fixed trial abort window and reward plateau parameters, and eliminating the blue light that signaled reward delivery. Mice were deemed task experts when they exhibited stable performance at $d' > 1$ (mean running d-prime) for three consecutive sessions. Expert mice performed 500 - 1000 trials daily.

**Task structure.** Task structure was identical to Ramamurthy et al., 2023[42]. Briefly, on each trial, a whisker stimulus was applied to either one randomly selected whisker (Go trials, 50–60% of trials) or no whisker (NoGo trials, 40–50% of trials). The whisker stimulus consisted of a train of five deflections separated by 100 ms. Every deflection was a rostrocaudal ramp-return movement (5 ms rise-fall time, 10 ms total duration, 25–300 µm amplitude, applied 5 mm from the face. NoGo trials presented the same stimulus on a dummy piezo that did not contact a whisker, so that any unmasked auditory cue from piezo movement was matched between Go and NoGo trials. Trial onset was irregular with an ITI of $3 \pm 2$ s. Mice had to restrict licking to greater than 3 s interlick interval (ILI) to initiate the next trial. On Go trials, mice were rewarded for licking within the 2.0 s response window with ILI of < 300 ms. Licking was not rewarded on NoGo trials. Each trial outcome was recorded as a Hit, Miss, False Alarm, or Correct Rejection.

Different delay periods were used for mice in different experiments (Supplementary Fig. S1C, D). Mice used in imaging or extracellular recording experiments had a delay period of 500 ms (for all spike recording mice and 7/14 2p imaging mice) or 1000 ms (for the other seven 2p imaging mice). This was used to temporally separate sensory-driven neural activity from action- and reward-related activity. Three mice used only for behavioral data collection were tested without a delay period, which enabled testing of attention effects on

lick response latency (Supplementary Fig. S1E). Different whisker deflection amplitudes were interleaved in each session, in order to explore attentional modulation as a function of stimulus amplitude (Supplementary Fig. S1F). Across all behavioral experiments, the proportion of Go trials at each amplitude were: 0–100 μm: 5.2%, 100–200 μm: 25.5%, and 200–300 μm: 69.3%. Across all imaging experiments, the proportions were 0–100 μm: 0.5%, 100–200 μm: 15.9%, 200–300 μm: 83.7%. Because trial history modulated detection performance at all stimulus amplitudes (Supplementary Fig. S1F), Go trials of all amplitudes were averaged to calculate mean behavioral and neural modulation as a function of trial history.

Because reward size varied with lick time in the response window, reward volume varied across trials. In addition, for mice with the 1000 ms delay period, mice sometimes licked on Go trials after stimulus presentation but before the response window opened, and thus earned no reward. These represent unrewarded Hits, so that attention effects could be quantified based on the absence of reward and reward size on prior Hit (Fig. 1F).

**Task variations.** We trained mice on three variations of the task, schematized in Supplementary Fig. S1G. All mice were initially trained on an "equal probability" (EqP) version of the task in which whisker identity on each Go trial was randomly selected from nine possible whiskers with equal 1/9 probability (EqP). The two other task variations were performed in a subset of sessions in some mice, and manipulated either the global or local probability of each specific whisker, while still randomly selecting whisker identity on each trial. In high probability (HiP) sessions, we manipulated the global stimulus probability of each whisker by presenting one whisker with a higher probability (80% of Go trials) than the others. This was done in 300–400 trial blocks, interleaved with standard EqP blocks on the same day. In high probability of same whisker (HiPSame) sessions, we manipulated the local probability of repeating the same whisker stimulus on consecutive trials, while maintaining the overall probability of each whisker at 1/9. This was done in blocks interleaved with EqP blocks.

All mice were tested on the EqP version, and either the HiP/EqP block design (13 mice) or the HiPSame/EqP block design (7 mice). Whisker-specific attentional cueing was observed in all 3 task variants (Supplementary Fig. S1H), so data from all variations were combined for the rest of the analyses.

A modified task version was used for extracellular recording experiments, in order to adequately sample trial history conditions when only 2–4 days of acute recording were possible per mouse. To do this, we reduced the number of whiskers sampled during Go trials from 9 whiskers to either 4 or 5 whiskers 5 (4-5-whisker task; Fig. 7A). The 4-5-whisker task was used in 3 mice for extracellular recordings, and was also applied in 2 mice that were used in PYR cell imaging, where performance could be compared to the standard 9-whisker task (Supplementary Fig. S7A).

In all these task variations, the presentation of Go or NoGo trials, and the identity of the whisker on each Go trial, were chosen randomly for each trial and was not contingent on the mouse's performance. The only exception was during a transient phase of training (Stage 6), when we briefly capped the maximum number of consecutive Go or NoGo trials to help ensure mice did not alter their response strategy as they were learning the delay period.

**Behavioral movies & DeepLabCut tracking**
Behavioral movies were acquired at 15–30 frames/sec using either a Logitech HD Pro Webcam C920 (modified for IR detection) or FLIR Blackfly S (BFS-U3-63S4M; used in video analyses). DeepLabCut[29] was used to track spontaneous face and body movements. Movies were manually labeled to generate training datasets for tracking facial motion (snout tip, whisker pad and 2-3 whiskers), body motion (corner of the mouse stage, whose motion reflects limb and postural

movements), pupil size (8 labels on the circumference of the pupil), eyelids (8 labels on the circumference of the eyelid), and licking (tongue and lickport). Three separate networks were trained (100,000-200,000 iterations) such that a good fit to training data was achieved (loss < 0.005). One network each was trained for face/body motion for the two camera setups (version 1 network: 1110 labeled frames from 37 video clips across 6 mice) and another for pupillometry (version 2 network: 2463 labeled frames from 27 video clips across 2 mice).

Behavioral movies from 74 sessions in 9 mice were analyzed for whisker motion (average across all whisker-related labels), body motion (stage corner) and pupil size (ellipse fit to the pupil markers). Blinking artifacts were removed using a one-dimensional moving median filter (40 frame window) applied to the trace of pupil size as a function of time. Pupil size measured on each frame was normalized to the mean pupil size over each individual session.

**Whisker paralysis by Botox injection**
Four mice (2 VIP-Cre;Ai162D mice and 2 Drd3-Cre;Ai162D mice) that were used for imaging experiments also underwent Botox injection to induce paralysis of whisking[42,85–87]. Both whisker pads were injected with Botox (Botulinum Neurotoxin Type A from Clostridium botulinum, List Labs #130B). A stock solution of 40 ng/μl Botox was prepared with 1 mg/ml bovine serum albumin in distilled water. Each whisker pad was injected with 1 μl of a 10 pg/μl final dilution using a microliter syringe (Hamilton). Whisking stopped within 1 day and gradually recovered in ~1 week. Following the initial dose, a 50% Botox supplement was injected once per week, as needed. Imaging was performed > 24 h after any Botox injection.

**Behavioral analysis**
Behavioral performance was assessed using the signal detection theory measures[88] of detection sensitivity ($d'$) and criterion ($c$), calculated from Hit rates (HR) and False Alarm rates (FA), as per their standard definitions:

$$d' = Z_{HR} - Z_{FA} \tag{1}$$

$$c = \frac{1}{2}\left(Z_{HR} + Z_{FA}\right) \tag{2}$$

where Z is the inverse cumulative of the normal distribution.

For each session, we first used a sliding d-prime to identify poor-performance periods at the beginning and end of the session that reflect satiety effects, and to exclude them from further analysis. To do this, $d'$ was computed across trials over the entire behavioral session. A sliding $d'$ cutoff (calculated over a 50 trial sliding window) was applied to the start and end of the session, and analysis was restricted between the first and last trial that met the threshold, which was termed the 'analysis period'. A standard sliding $d'$ cutoff of 0.5 was used for all behavioral and imaging analyses. We tested $d'$ cutoffs 0.5, 0.7 and 1 to ensure that the choice of $d'$ cutoff did not affect key results. A sliding $d'$ cutoff of 1.2 was used for extracellular recording analyses.

Next, we analyzed trial history effects within the analysis period of each session. To calculate d-prime for a specific trial history, we (1) identified all Go trials with that trial history in the analysis period, and all NoGo trials with the same trial history in the analysis period, and then (2) calculated d-prime from the Hit rate and FA rate over this entire set of Go and NoGo trials. For example, to calculate d-prime for 'prior trial = NoGo', we calculated the Hit rate for all Go trials that had ≥1 prior NoGo during the entire analysis period, and FA rate for all NoGo trials that had ≥1 prior NoGo during the entire analysis period. Likewise, to calculate d-prime for 'prior 1 Hit same', we calculated Hit rate for all Go trials that had 1 prior Hit to the same whisker, and FA rate for all NoGo trials with 1 prior Hit (to any whisker). This method

enables us to measure how recent trial history influences the average d-prime (by averaging across all trials with that history, spread throughout the session).

## Definition of trial histories

For each *current trial*, trial history conditions were defined based on outcome and stimulus on *prior trials*, as follows. These definitions are illustrated in Figure. S1B.

History conditions were defined for the *current Go trials* as follows:

1. Prior Miss
   a. On the *same* whisker: The current trial is a Go, and the outcome on the previous trial was a Miss to the same whisker.
   b. On a *different* whisker: The current trial is a Go, and the outcome on the previous trial was a Miss to a different whisker.
2. Prior NoGo: The previous trial was a NoGo (any outcome).
3. Prior 1 Hit
   a. On the *same* whisker: The current trial is a Go, and the outcome on the previous trial was a Hit to the same whisker.
   b. On a *different* whisker: The current trial is a Go, and the outcome on the previous trial was a Hit to a different whisker.
4. Prior > 1 Hit:
   a. On the *same* whisker: The current trial is a Go, and the outcomes on the previous two or more trials were Hits to the same whisker as the whisker presented on the current trial.
   b. On a *different* whisker: The current trial is a Go and the outcomes on the previous two or more trials were Hits to a single consistent whisker that was a different identity than the whisker presented on the current trial (i.e., two or more Hits in a row to the same whisker that differed from the current Go trial).

For each history condition defined above for *current Go trials*, a matched condition was defined for *current NoGo trials*. This allowed us to compute behavioral *d′* and *c* values within each history condition, and to compare neural signals on Go and NoGo trials within each history condition. History conditions for *current NoGo trials* were defined as follows:

1. Prior Miss: The current trial is a NoGo, and the outcome on the previous trial was a miss to any whisker. NoGo trials in this category were used for comparison with Go trials in both categories 1a and 1b above.
2. Prior NoGo: The previous trial was a NoGo (any outcome).
3. Prior 1 Hit: The current trial is a NoGo, and the outcome on the previous trial was a hit to any whisker. NoGo trials in this category were used for comparison with Go trials in both categories 3a and 3b above.
4. Prior > 1 Hit: The current trial is a NoGo, and the outcomes on the previous two or more trials were Hits to any single repeated whisker. NoGo trials in this category were used for comparison with Go trials in categories 4a and 4b, above.

The history conditions above were defined based on sequences of consecutive trials, including both Go and NoGo. The history-dependent effects on detection behavior were maintained when history conditions were defined by ignoring NoGo trials and categorizing history based solely on Go trials. NoGo trials were ignored when characterizing the temporal profile of attention as a function of time since last Go trial (Fig. 2B).

## 2-photon calcium imaging

A Sutter Moveable-Objective Microscope with resonant-galvo scanning (RESSCAN-MOM, Sutter) was used to perform 2p imaging in expert mice. A Ti-Sapphire femtosecond pulsed laser (Coherent Chameleon Ultra II) tuned to 920 nm, or an ALCOR 920 nm fixed

wavelength femtosecond fiber laser (Spark Lasers), was used for GCaMP6s excitation. A water-dipping objective (16x, 0.8 NA, Nikon) was used, and emission was band-pass filtered (HQ 575/50 filter, Chroma) and detected by GaAsP photomultiplier tubes (H10770PA-40, Hamamatsu). Single Z-plane images (512 × 512 pixels) were acquired serially at 7.5 Hz (30 Hz averaged every 4 frames) using ScanImage 5 software (Vidrio Technologies). Laser power measured at the objective was 60–90 mW. On average, 14 imaging fields (305 μm x 305 μm) were obtained per mouse at depths of 110–250 μm below the cortical surface[89]. If there was > 25% XY overlap, imaging fields were required to be at least 20 μm apart in depth to avoid repeated imaging of the same cells. After completion of all imaging experiments, the mouse was euthanized, and the brain was collected to perform histology.

## Histological localization of imaging fields

The brain was extracted and fixed overnight in 4% paraformaldehyde. After flattening, the cortex was sunk in 30% sucrose and sectioned at 50–60 μm parallel to the surface. Cytochrome oxidase (CO) staining showed surface vasculature in the most superficial tangential section as well as boundaries of barrels in L4. Histological sections were manually aligned using Fiji (https://imagej.net/software/fiji/)[90], and imaging fields were localized in the whisker map, aided by the surface blood vessels imaged at the beginning of each session. The centroid of each of the nine anatomical barrels corresponding to whiskers stimulated in each session and the XY coordinates of all imaged cells were localized relative to barrel boundaries. A cell was located within a specific barrel column if > 50% of its pixels were within its boundaries, and cells outside barrel boundaries were classified as septal cells. Major and minor axes of all barrels were averaged to calculate the mean barrel width.

## Image processing, ROI selection, and ΔF/F calculation

Custom MATLAB pipeline code (Ramamurthy et al., 2023[42]; adapted from LeMessurier, 2019[91]) was used for image processing. Correction for slow XY drift was performed using dftregistration[92,93]. Regions-of-interest (ROIs) were manually drawn as ellipsoid regions over the somata of neurons visible in the average projection across the full imaging movie after registration. The mean fluorescence of the pixels in each ROI was calculated to obtain the raw fluorescence time series. For PYR cell imaging, neuropil masks were created as 10 pixel-wide rings beginning two pixels from the somatic ROI, excluding any pixels correlated with any somatic ROI (r > 0.2). Mean fluorescence of neuropil masks was scaled by 0.3 and subtracted from raw somatic ROI fluorescence. Neuropil subtraction was not performed for VIP cells, which were spatially well-separated. The mean fluorescence time series was converted to ΔF/F for each ROI, defined as $(F_t-F_0)/F_0$, where $F_0$ is the 20th percentile of fluorescence across the entire imaging movie and $F_t$ is the fluorescence on each frame.

## Quantification of whisker-evoked responses

Whisker-evoked ΔF/F signal on Go trials was quantified in a post-stimulus analysis window (7 frames, 0.799 s), relative to pre-event baseline window (2 frames, 0.270 s). Whisker responses were measured as (mean ΔF/F in the post-stimulus window – mean ΔF/F in the baseline window) for each ROI. On NoGo trials, the ΔF/F analysis was aligned to the NoGo stimulus (dummy piezo deflection). Whisker responses for each cell were normalized by z-scoring to prestimulus baseline activity. For some analyses, each ROI's Go-NoGo response magnitude to every whisker was also calculated as (median whisker-evoked ΔF/F signal across Go trials – median ΔF/F signal across NoGo trials). Trials aborted due to licks occurring during the post-stimulus window (0–0.799 s) were excluded from analyses. If at least one whisker produced a significant response above baseline activity (permutation test), the cell was considered to be whisker-responsive. This

was done by combining the whisker-evoked ΔF/F signal distribution on Go trials with the ΔF/F signal distribution on NoGo trials, randomly splitting the combined distribution into two groups and comparing the difference in their means to the true Go-NoGo distribution difference (10,000 iterations). Differences greater than the 95th percentile of the permuted distribution were assessed as significant. The nine whisker response p-values were corrected for multiple comparisons (False Discovery Rate correction[94]). Cells without a positive ΔF/F response to at least one whisker were considered non-responsive.

The standard method for assessing whisker-responsiveness used all trials belonging to each session (combining trials across all history conditions). In the analysis of attentional modulation of receptive fields, the significance of whisker responses was separately assessed using only trials in the Prior NoGo category and compared to trials in Prior >1 Hit condition, which allowed us to test whether there was history-dependent acquisition of whisker-evoked responses by previously non-responsive cells.

### Definition of each cell's columnar whisker (CW) and best whisker (BW)

Each cell's anatomical home column was determined by histological localization of the cell relative to barrel column boundaries. For cells located within column boundaries, the CW was the whisker corresponding to its anatomical home location. For septa-related cells (i.e., those outside of column boundaries and above an L4 septum), the CW was the whisker corresponding to the nearest barrel column. The best whisker (BW) was defined for each cell as the whisker that evoked the numerically highest magnitude response.

### Attention modulation index (AMI)

Multiple AMI metrics were used to quantify attentional modulation of whisker response magnitude in individual cells. The definitions were:

AMI ( >1Hit-NoGo):

$$AMI_{>1HitSame-NoGo} = \frac{Go_{Prior>1HitSame} - Go_{PriorNoGo}}{|Go_{Prior>1HitSame} + Go_{PriorNoGo}|} \quad (3)$$

$$AMI_{>1HitDiff-NoGo} = \frac{Go_{Prior>1HitDiff} - Go_{PriorNoGo}}{|Go_{Prior>1HitDiff} + Go_{PriorNoGo}|} \quad (4)$$

AMI( >1HitSame->1HitDiff):

$$AMI_{>1HitSame->1HitDiff} = \frac{Go_{Prior>1HitSame} - Go_{Prior>1HitDiff}}{|Go_{Prior>1HitSame} + Go_{Prior>1HitDiff}|} \quad (5)$$

where Go = mean whisker-evoked ΔF/F for current Go trials (on any whisker) with the specified trial history.

### Attentional modulation of receptive fields

Population average 9-whisker receptive fields were constructed centered on the CW, and included the CW plus the 8 immediately adjacent whiskers. A separate population average receptive field was calculated for each trial history (Fig. 5A). These represent the average tuning of cells within each whisker column, following each trial history. To test for shifts in receptive fields by attention to specific whiskers, we first computed the center of mass (CoM) of the population average receptive fields for each history condition. CoM was calculated in a Cartesian CW-centered whisker space, as defined in Supplementary Fig. S5B. The CW position is considered the origin in this space. Receptive field shifts associated with prior trial history were visualized as vectors from CoM measured after NoGo trials, to CoM measured after >1 prior hit to specific whiskers.

To quantify the receptive field shift (ΔRF CoM) for individual cells, we defined the *attention axis* as the axis connecting the CW position to the attended whisker position in the Cartesian CoM space. We

projected the CoM_PriorNoGo and CoM_Prior>1Hit onto this axis, and computed the receptive field shift (ΔRF CoM) as the distance between these projected positions normalized to the distance from CW to attended whisker along the attention axis (Supplementary Fig. S5C). Since not all whisker positions could be sampled for all cells across history conditions, ΔRF CoM was quantified only for the subset of cells for which at least 6 of the 9 whisker positions were sampled in both Prior >1 Hit and Prior NoGo conditions. Only response magnitudes at whisker positions sampled in both Prior >1 Hit and Prior NoGo conditions for any given cell contributed to the CoMs computed for that cell. The RF shift was computed separately for each attended whisker position that was sampled for a given cell, and then averaged across these attended whisker positions to generate a single RF shift metric for that cell. To quantify the effects of attending to the CW on CW and SW responses in individual cells, we calculated a response modulation index (RMI) separately for CW responses and SW responses, defined as:

RMI ( >1Hit-NoGo):

$$CW\,RMI_{>1HitCW-NoGo} = \frac{CW_{Prior>1HitCW} - CW_{PriorNoGo}}{|CW_{Prior>1HitCW} + CW_{PriorNoGo}|} \quad (6)$$

$$SW\,RMI_{>1HitCW-NoGo} = \frac{SW_{Prior>1HitCW} - SW_{PriorNoGo}}{|SW_{Prior>1HitCW} + SW_{PriorNoGo}|} \quad (7)$$

where CW = mean whisker-evoked ΔF/F for current CW trials with the specified trial history, and SW = mean whisker-evoked ΔF/F for current SW trials with the specified trial history, where SW is any of the 3 SWs with the strongest magnitude responses for that cell.

Columnar whisker preference index (Fig. 5F) was calculated as:

$$CW\,preference(Prior>1HitCW) = \frac{CW_{Prior>1HitCW} - SW_{Prior>1HitCW}}{|CW_{Prior>1HitCW} + SW_{Prior>1HitCW}|} \quad (8)$$

$$CW\,preference(PriorNoGo) = \frac{CW_{PriorNoGo} - SW_{PriorNoGo}}{|CW_{PriorNoGo} + SW_{PriorNoGo}|} \quad (9)$$

where CW and SW are defined as for RMI.

### Somatotopic profile of attentional modulation

To quantify the somatotopic profile of attentional modulation in S1 (Fig. 4K), we considered each of the 9 tested whiskers separately. For each whisker (termed the reference whisker), every cell was placed in a spatial bin representing its distance to the center of the reference whisker column in S1. Both columnar and septal-related cells were included. Mean whisker response magnitude was calculated, separated by trial history, in each bin. This was repeated for all 9 reference whiskers, and Fig. 4K shows the average response. Thus, the Prior NoGo trace reflects normal somatotopy, i.e., the normal point representation of an average whisker. The somatotopic profile of attentional modulation is evident as the difference between other history conditions and the Prior NoGo condition.

A similar binning procedure was used to calculate the somatotopic profile of AMI modulation across S1 columns (Supplementary Fig. S4A).

### Imaging analysis for VIP cells

Analysis of VIP cell responses was performed similarly to PYR cells, except that neuropil subtraction was not performed. To separate whisker-evoked VIP responses from slow trends in VIP baseline activity related to whisker motion, body motion and arousal in the ITI (Ramamurthy et al., 2023[42] and Fig. 3), we applied linear baseline detrending. For detrending, the median pre-stimulus baseline trace (in a 1.07 s window) was calculated across Go trials (aligned to stimulus

onset time) and NoGo trials (aligned to dummy piezo onset time). A line was fit to this median trace. This line was extrapolated and subtracted from each individual trial to yield the full peri-stimulus trace. This linear detrending was done separately for each history condition, due to the differences in pre-stimulus slopes for each condition. Note that linear detrending for prior same and prior different categories in each reward condition was identical. Analysis of history effects on VIP whisker responses (Fig. 8F, G) was performed after linear detrending. While VIP cells did not show whisker-specific attentional modulation (Fig. 7F, G), we verified that PYR cells still showed whisker-specific attentional boosting after detrending with the same methods.

### N's for PYR and VIP cell analysis

The full PYR imaging dataset consisted of 6906 PYR cells, and the full VIP imaging dataset consisted of 1843 VIP cells, that were imaged in sessions that met all stated criteria for data inclusion. These full datasets were used to generate the population activity traces in Fig. 4D and Fig. 8D, E, respectively. Within each of these datasets, cells varied in having a large number of trials for some history conditions, but could have too few trials for meaningful analysis in other history conditions (in which case they were omitted from population analysis of those conditions). For single-cell analysis, we identified a subset of these cells in which every history condition was sampled sufficiently to calculate a meaningful single-cell ΔF/F trace for each history condition, and to calculated attention metrics (AMI) on the single-cell level. This was 2115 PYR cells and 852 VIP cells, which were used to for the AMI analyses (Figs. 4H, I, 8G, H) and to show ΔF/F traces for each history condition for individual cells (Supplementary Figs. S3G and S8). We verified that attentional effects on population average ΔF/F traces shown with the full dataset (Figs. 4D and 7D, E) were also observed, virtually identically, for this subset of cells that were sampled in all history conditions (Supplementary Fig. S3G).

### Neural decoding from population activity on single trials

We built a neural decoder that predicts the presence or absence of a whisker stimulus (i.e., whisker detection) from single-trial neural activity of all individual whisker-responsive L2/3 PYR cells in each 2p imaging session. Activity was defined as mean ΔF/F in the post-stimulus window (0–0.799 s) in each trial. We built a generalized linear model ('cvglmnet' in Matlab)[95] with weights fit using ridge regression (L2-penalized logistic regression, alpha = 0; optimal lambda estimated for each session using 10-fold cross-validation) to predict the presence or absence of a whisker stimulus on each trial from single-trial neural activity (ΔF/F) of individual ROIs imaged in each session. A separate decoder was fit for each session. Decoder performance was tested on held-out trials using k-fold cross-validation with 10 randomly assigned folds.

Training/testing datasets were randomly re-sampled (majority class undersampled to match the minority class) to have identical numbers of trials within each response category (Hit, Miss, CR, FA), in order to remove bias. Decoder performance was measured as the average fraction of trials classified as containing a whisker stimulus (assessed over 25–50 iterations) and compared to performance for a decoder trained with shuffled trial labels.

For each field, we defined the fBW (field best whisker) as the whisker which evoked the numerically highest mean population ΔF/F. A single decoder was trained for each session to predict any whisker from training data containing Go trials from all whiskers, as well as NoGo trials. Decoder performance was assessed either for detecting any whisker, or just the fBW, or just non-fBW trials.

### Extracellular recordings

For surgical preparation for mice used in extracellular recording experiments, methods were similar to that described above, except a lightweight chronic head post was affixed to the skull using cyanoacrylate glue and Metabond, and ISOI was performed to localize D-row (D1, D2, D3) barrel columns in S1. A 5-mm diameter glass coverslip (#1 thickness, CS-3 R, Warner Instruments) was placed over the skull, sealed with Kwik-Cast silicone adhesive (World Precision Instruments) and dental cement. Mice recovered for 7 days prior to the start of behavioral training (~ 4 weeks before recording).

The day before recording, mice were anaesthetized with isoflurane, the protective coverslip was removed, and a craniotomy (~ 1.2 × 1.2 mm) was made over S1, centered over the D-row (D1, D2, D3) barrel columns localized by ISOI. A plastic ring was cemented around the craniotomy to create a recording chamber. During recording sessions, mice were anesthetized and positioned on the rig. A reference ground was attached inside the chamber. The craniotomy and reference wire were covered in a saline bath. Recordings were made with Neuropixels 1.0 probes using SpikeGLX software release v.20201024 (http://billkarsh.github.io/SpikeGLX/)[96], Imec phase30 v3.31. Acute recordings were made in external reference mode with action potentials (AP) sampled at 30 kHz at 500 x gain, and local field potential (LFP) sampled at 2.5 kHz at 250 x gain. The AP band was common average referenced and band-pass filtered from 0.3 kHz to 6 kHz. The Neuropixels probe was mounted on a motorized stereotaxic micromanipulator (MP-285, Sutter Instruments) and advanced through the dura mater (except in cases where the dura had detached during the craniotomy). To reduce insertion-related mechanical tissue damage and to increase the single-unit yield, the probe was lowered with a slow insertion speed of 1–2 µm/sec. The probe was first lowered to 700 µm, and a short 10 min recording was conducted to map its location in S1. After identifying the columnar whisker for the recording penetration location, probe insertion continued until the final depth was reached. The probe was then left untouched for ~20 min. The craniotomy was sealed after probe insertion with silicone sealant (Kwik-Cast, World Precision Instruments) to prevent drying. Anesthesia was then discontinued, and mice were allowed to fully wake up before recording.

After recording was complete for the day, the probe was removed, the craniotomy was sealed with silicone sealant (Kwik-Cast, World Precision Instruments), and the recording chamber was sealed with a cover glass and a thin layer of dental cement. 3-4 sequential days of recording were performed in each mouse, with the probe located in a different whisker column in S1 on each day. On the final recording day, a Neuropixels probe was coated with red-fluorescent DiI (1,1'-Dioctadecyl-3,3,3',3'-tetramethylindocarbocyanine perchlorate; Sigma-Aldrich) dissolved in 100% ethanol, 1-2 mg/mL, which was allowed to partially dry on the probe before probe insertion. The probe was briefly inserted to deposit DiI at the recording and several fiducial sites. The mouse was euthanized and the brain was extracted, sectioned tangentially to the pial surface, and processed to stain for cytochrome oxidase (CO). DiI deposition sites in L2/3 were localized relative to column boundaries in CO from L4 (Supplementary Fig. S7B).

The columnar location of each recording penetration was determined from DiI marks on the last recording day, plus relative locations of other penetrations based on reference images of surface vasculature and microdrive coordinates. The laminar depth of each recording was determined from current source density (CSD) and LFP power spectrum analysis, as described below.

### Analysis of extracellular recording data

**Spike Sorting.** Spike sorting was performed by automatic clustering using Kilosort3[97] followed by manual curation using the 'phy' GUI (https://github.com/kwikteam/phy)[98]. Isolated units were manually inspected for mean spike waveform, stability over time, and inter-spike interval refractory period violations (we required that <2% of intervals <1.5 ms). Only well-isolated single

units were analyzed. Single units were classified as regular-spiking or fast-spiking based on trough-to-peak duration of the spike waveform at the highest-amplitude recording channel, with a separation criterion of 0.45 ms[99] (Supplementary Fig. S7C). Only data from regular spiking units was analyzed here.

**Layer assignment using CSD and LFP power spectrum analysis.** To calculate the CSD for each recording penetration, the whisker stimulus-evoked local field potential (LFP, 500 Hz low-pass) was calculated for each Neuropixels channel. LFP traces were normalized to correct for variations in channel impedance and were interpolated between channels (20 μm site spacing, 1.6-2x interpolation) prior to calculating the second spatial derivative, which defines the CSD[100]. For visualization, CSDs were convolved with a 2D (depth x time) Gaussian, which revealed depth-restricted regions of current sources and sinks in response to each whisker stimulus. The L4-L5A boundary was defined from CSD as the zero-crossing between the most negative current sink (putative L4) and the next deeper current source (putative L5A) (Supplementary Fig. S7F). To estimate the brain surface location (defining the top of L1), we computed the LFP power spectrum as a function of channel depth ('lfpBandPower', https://github.com/cortex-lab/neuropixels/)[101]. A sharp increase in low-frequency LFP power marked the brain surface, which was used to verify the correct selection of the L4-L5A boundary from CSD. Each cortical layer was then assigned boundary depths based on layer thicknesses reported in Lefort et al., 2009[102].

**Whisker response quantification and attentional modulation for spike recordings**

Firing rates for Go and NoGo trials were quantified in a 0.5 s window after stimulus onset (lick-free window). Units were classified as whisker-responsive or non-responsive by testing for greater firing rate on Go vs NoGo trials, using a permutation test, as for the 2p imaging data. To quantify whisker-evoked response magnitude, the firing rate for each unit was z-scored relative to the pre-stimulus baseline. Peristimulus time histograms (PSTHs) were constructed using 10 ms bins aligned to stimulus onset. AMI metrics were used to quantify attentional modulation of whisker-evoked spiking and were calculated exactly as for 2p imaging data, but from z-scored whisker-evoked firing rate in the post-stimulus window.

**Statistics and reproducibility**

Statistics were performed in MATLAB. Sample size (n) and p-value for each analysis are reported in the figure panel, with the statistical test reported in the figure legend or in *Results*. All statistical tests were two-sided unless stated otherwise stated. Two-sample permutation tests for the difference in means were used to assess differences between groups (referred to as 'permutation tests' throughout), and one-sample permutation tests were used to assess paired differences relative to zero. A binomial exact test was used to assess the statistical significance of deviations from the expected distribution of observations into two categories. Summary data are reported as mean ± SEM, unless otherwise specified. Population means were compared using permutation tests, using False Discovery Rate (FDR) correction[94] for multiple comparisons as needed. To compare population means with multiple subgroups, we first assessed whether a main effect was present using a permutation test, and then performed post hoc tests for pairwise means, correcting for multiple comparisons (with FDR correction). We used a significance level (alpha) of 0.05. All statistical tests reported in the *Results* use n of cells, but we also verified that all major behavioral and neural effects were consistent across individual mice. We show individual mouse data

for comparison with population data summaries. To account for inter-individual variability, we also verified all key results using linear mixed effects models with the formula:

$$\text{Response Variable} \sim \text{History} + \text{Mouse Sex} \\ + (1|\text{Mouse ID}) + (1|\text{Session ID}) \quad (10)$$

In which history and mouse sex are included as fixed effects, and mouse ID and session ID are modeled as random effects.

**Reporting summary**

Further information on research design is available in the Nature Portfolio Reporting Summary linked to this article.

## Data availability

The behavior data, 2p imaging data and extracellular recording data generated in this study have been deposited in the Zenodo repository[103], under accession code https://doi.org/10.5281/zenodo.14888799.

## Code availability

Original code for data analysis is available in the Zenodo repository, https://doi.org/10.5281/zenodo.14888799.

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

## Acknowledgements

This work was supported by NIH Grant R01 NS092367 (D.E.F.) and NIH Grant 5 K99 NS129753-02 (D.L.R.).

## Author contributions

D.L.R. and D.E.F. designed the study. D.L.R., L.R., C.C., S.L., and A.C. performed the experiments. D.L.R., L.R., C.C., and S.L. analyzed the data. D.L.R., L.R., and D.E.F. wrote the manuscript.

## Competing interests

The authors declare no competing interests.
