## [Transparent Peer Review file · Nature Communications]

Reward history guides focal attention in whisker somatosensory cortex

Corresponding Author: Dr Daniel Feldman

Version 0:

Reviewer comments:

Reviewer #1

(Remarks to the Author)

This is an excellent and exciting study from Daniel Feldman's lab. Over the past several years, Feldman's lab has been exploring different aspects of spatiotemporal coding in the mouse whisker system revealing complex interactions between patterns of multi-whisker sensory inputs and receptive fields in the barrel cortex. In this new study, the authors designed a clever behavioral paradigm and analyses to study the impact of trial/reward history on spatial attention in a whisker detection task. By randomly presenting the whisker stimulus to one of nine individual whiskers, the authors show that consecutive successful detections (Hit trials) of the whisker stimulus applied to the same whisker result in increased detectability if the next stimulus is applied again to that same whisker, but result in a slight decrease in detectability if the stimulus is applied to a different whisker. This shift in spatial attention to the last rewarded whisker cannot be explained by a difference in motor activity or general arousal, and it is correlated with an increased responsiveness of cortical neurons responding to the attended whisker. This response enhancement was more prominent in neurons in the surrounding barrel columns and resulted in a receptive field shift towards the attended whisker. The authors also investigate the possible contribution of VIP interneurons in mediating this trial-history-related change in neuronal response, finding a correlation between VIP neuron activity and general motor activity and arousal, but not with the whisker-specific shift in attention. Overall, the results are very compelling and exciting, and I have only a few questions and suggestions for clarification:

- 1) all along the study the authors use different versions of the task, using no delay, a 0.5 s or a 1.0 s delay, using different stimulus amplitudes (figure 1F and S1F). It is not always easy to understand which sessions (i.e. version of the task) are used for the different analyses. The authors should try to clarify this point whenever appropriate.
- 2) In the same line, the use of different stimulus amplitude is briefly mentioned in the Results section but is not described in the Methods section. The authors should describe more clearly when stimuli of different amplitudes are used and the respective proportion of trials for each amplitude. The authors could also show a psychometric curve in the supplementary information.
- 3) Lastly, if whisker stimuli of different amplitudes are always used, how is this accounted for in the analyses? Are all trials used, regardless of stimulus amplitude, to calculate d' or neuronal evoked responses? Are stimulus amplitudes accounted for in the trial history?
- 4) On another topic, the authors should show longer traces for evoked responses (e.g. Figure 3D). The baseline in particular is very short. For PYR L2/3 neurons, there seems to be a positive trend in Nogo trials on some traces. It might be interesting to consider a detrending approach similar to the one used for VIP neurons in Figure 7E?
- 5) Regarding the quantification of movement in Figure 2A, I fail to understand the reason for subtracting the baseline activity 0.25 s before the onset of the whisker stimulus. It seems to me that it would be interesting to know if there is a difference in movement depending on trial history when the whisker stimulus is presented?
- 6) In Figure 4A, why not show the change in evoked response for consecutive hits for the CW whisker (central gray panel)? I believe it is shown in a separate figure in S5A?
- 7) For the decoder part of Figure 5, my understanding is that the detection of the whisker stimulus is based solely on the

average activity of the population for a given trial. I wonder whether a decoder that better takes into account the complex dynamics of the population – that is, the activity of each individual neuron – might not achieve better performance. In particular, I am a little surprised by the decoder's high level of "false alarms" on CR trials (nearly 40%) and the low level of correct detection on Miss trials (less than 50%).

Sylvain Crochet
(Please do leave my name apparent to the authors)

Reviewer #2

(Remarks to the Author)

This very interesting paper investigates how prior reward history influences behavioral performance and neuronal responses in mouse S1. Using a Go/NoGo whisker deflection detection task, the authors demonstrate that recent associations of a reward with a whisker stimulus leads to the enhanced detection probability of the next whisker stimulus. The authors attribute this to an enhancement of spatially selective attention. They then use 2-photon imaging and spike recordings to show that whisker-evoked L2/3 and L5 pyramidal neural responses are also enhanced in the corresponding cortical columns if that whisker had been stimulated in prior trial and resulted in a reward. They then show that VIP interneurons, whose activity is often associated with arousal and attention, are not modulated by the reward history but instead respond to arousal and global behavioral state. This reward history-driven response modulation offers an interesting novel framework for studying behaviorally relevant sensory processing in cerebral cortex.

This is a very interesting study both from the perspective of reward history-associated perception, as well as for our understanding of sensory processing in cortex. I applaud the thoroughness of this study, the various angles of analyses as well as the depth at which the authors have gone through the data. At a first read, I found all aspects and findings intuitive and convincing. Upon a second and third read I found myself getting somewhat confused by a couple of points, which I am sure the authors will be able to clarify or correct.

1. The main issue that has caused confusion is the use of d-prime (d') to indicate the instantaneous behavioral performance on a particular trial (which they term 'current' trial, e.g. from Fig. 1D onward). In my understanding, and as the authors describe in the methods, d' is derived from a series of 50 trials around the 'current trial'. This makes sense if one were interested in the general slowly evolving fluctuations in behavioral performance (i.e. signal sensitivity over a series of trials, such as in Fig. S1A). But I fail to see the relevance of this measure for estimating the effect of reward history (i.e. prior Go/hits). Since d' is based on both hit rates (HR) as well as false alarm rates (FR) over many trials, the measure is not a clean indicator of the impact of the nearest prior trials. The 'probability of lick on current trial' with various priors (Fig. 1E) is in my opinion the only measure that cleanly indicates the effect of reward history.

Further corroborating my point here, I find the values of d' and the deltas thereof very high (2.45 for the >1 Go/hit same whisker cases in Fig. 1D), which I think implies that if mice have HR on those trials of around 0.85 (roughly estimated as indicated in Fig. 1E), their FR should be around 0.08. For >1 Go/hit on different whisker cases, the d' is 0.82, but the HR on those trials isn't that much different, about 0.7 (Fig. 1E). That means that it is mostly the FR that has gone up in these sessions (to about 0.4). I think this point is more less reflected in Fig. 1G: for the 'pior different' data points, it is not so much the HR, but rather the FR that goes up. I feel that this is not very informative of the 'attention' effects of the pior Go/hits. Remarkably, the FR in all NoGo trials of Fig. 1E is never at 0.4, but mostly 0.2; so, I hope the authors can understand my confusion.

The use of d' also includes the risk of intrinsic bias because the '> 1 hits' cases may include more Go trials than the other cases, and particular Go/hit trial sequences are probably used repeatedly to calculate d' for different 'current' trials. What happens in a series of many consecutive Go trials when the mouse is at the top of its performance? In these cases, d' prime will rely only on very few NoGo trials, and hence the FR will be low but very capricious.

I am sure the authors have thought deeply about this and will be able to provide solid arguments for using the d' measure. It just was not intuitive to me, and I therefore encourage them to introduce and support the methodology more elaborately.

2. Related to point 1, the authors mention in the methods that 'Mice were deemed task experts when they exhibited stable performance at $d' > 1$ '. However, the d' in expert mice was on average 0.923 which corresponds to an approximate fraction of correct trials of 0.6 -slightly above the chance level. Does this mean that the data presented in this study also pertain to mice that are still in the process of learning the association between any whisker stimulation and the reward. In the literature, mice are considered experts in Go/No-Go tasks when d' exceeds at least 1.5. Could the authors comment on that?

3. In figure 2A-B, the whisker motions for NoGo trials seems also sensitive to prior trial history. Is this associated with a higher FR? In figure 2B, the magnitude of the change of whisker position change is similar in all 3 trial types, including for NoGo trials. Can this be explained by an overall increase in compulsive licking behavior when mice experience several consecutive Hits on the same whisker?

4. Related to my previous point, the activity of L2/3 PYR neurons is higher in NoGo trials when it was preceded by multiple Hits (Figure 3D and E). As argued under point 3, could this mean that most of the NoGo trials in the 'Prior trial >1Hit' condition are FAs whereas most of NoGo trials in the 'Prior trial NoGo' condition are CRs? If one were to consider that the repetition of Go/hits on the same whisker could provoke a transiently more compulsive licking behavior, the interpretation of the data in this study would be very different. Can the authors exclude this? I would like to invite to comment on this in the

paper as well.

5. On line 134: "Trial history effects were driven by stimulus-reward association, not by stimulus salience or reward probability, because whisker deflections were physically identical, and reward probability was always 100% for each whisker." Shouldn't the authors report the probability of licking for all 9 whiskers to make this statement? Even though the whiskers' deflections are identical, the detection and perception of stimuli on different whiskers might substantially differ. In line 259, the authors state that they "observed modest, above-chance performance for detecting any of the 9 whiskers". This claim is made without statistically testing the performance against the chance level of 50%.

6. In figure 5, how do the authors explain that the fraction of trials predicted as stimulus is around 40% for the NoGo condition? I find this value particularly high given that no stimulus is delivered.

7. How do the authors explain, in Fig. 5E-G, that the fraction of trials detected as stimulus is not improved when there was 1 Hit on the same whisker previously?

8. In figure 7D, the activity traces of VIP neurons are somewhat puzzling. The authors state that the declining baseline is the result of prior trial activity. However, in these examples, the transient evoked by the whisker stimulation is relatively small and the average activity keeps declining, below baseline levels, about 1.5 s after the whisker stimulation. This must mean that there is a second large peak of activity occurring during the reward period. Moreover, I don't understand how multiple NoGo trials can produce a constantly declining baseline (dashed line of left examples). Can the authors comment on this, and perhaps they could show example traces that extend into the reward period and the next trial?

9. I think the authors make a convincing case for attention effects, but I am curious to hear if they considered short-term plasticity as a modulating process. If consecutive whisker stimuli plus a reward leave a short-lived (e.g. around 10 seconds) plasticity trace in the corresponding barrel column, one would expect a similar phenomenon, i.e. >1 Go/hit increases the pyramidal cell responses, which subsequently increases the detection probability. In this case the VIP neuron activity during the trial wouldn't be the modulating force but rather their activity in between the trials (i.e. at the rewards) – equivalent to an eligibility trace. If interesting, they could include this as a discussion point.

Minor points:

10. Figure 1D and other panels:

The design of the behavioral task is well explained, but the boundaries of the number of Go and NoGo sequences is not so clear (related to main point 1). Did the authors restrict the number of consecutive Go or NoGo trials?

11. For the reader to get a good intuition for the task, it might be helpful to move figure S1A to figure 1

12. Figure S1B: Above the D2 whisker trial following C2 (Miss), should the color perhaps be red instead of blue

Reviewer #3

(Remarks to the Author)

Ramamurthy et al present a novel somatosensory spatial attention task and explore its cellular mechanistic basis. By exploiting the history of which spatially localised stimuli were rewarded they show that mice can be encouraged to shift their attention between specific whiskers across strings of trials. Through a combination of behaviour, 2-photon calcium imaging and electrophysiological recordings, the authors present compelling and well controlled evidence for both behavioural and neural correlates of attention during their task. The authors do well to emphasise the distinction between the more commonly studied bottom-up or top-down attention and this phenomenon of attentional capture guided by reward history.

Additionally, they examine the activity of VIP interneurons, which have been hypothesised to be key in attentional control. The authors find that VIP interneurons do not carry an attention signal, corroborating recent work in the visual system.

Overall this is a very exciting and important study which both establishes a novel attentional paradigm and also rules out a key cellular hypothesis.

We have a few points which if addressed by the authors would make the study even stronger.

1. It would be interesting to know what effect the reward cued focal attention in this task has on inter-neuronal correlations and how it compares to the effects observed in previous studies of attention.

2. Is it possible that the spread in behavioural and neural effect of repeat stimulation onto the adjacent row and arc could in part be a result of the piezo array? That an active piezo subtly activates its neighbours?

3. There could be more emphasis in the text on the distinction between this phenomenon of attentional capture guided by reward history and bottom-up attention, particularly in the introduction section. Some discussion about how the neural boosting the authors have discovered compares to the existing literature on this type of attention.

Other minor points:

1. At some points, multiple nested samples are presented. For example, line 325 "(n = 7 mice, 103 sessions, 1843 VIP cells)" It would be helpful to know the range in sessions per mice and in VIP cells per session.

2. The wording when discussing the decoding of neural activity could be clearer, specifically about what is being decoded.

E.g. Lines 832-833 – is the decoder classifying which of the 9 whiskers was stimulated on a given trial? Or is it providing a binary answer of whisker stimulated vs no stimulation?

3. In supplementary figures S3E and S7 it's unclear what the dividing line in the columns is. The onset of whisker stimulation?

4. Why is an n of 1843 VIP cells stated in the text and Figure 7E, but only 852 cells in Figure 7G and Figure S7?

Reviewer #4

(Remarks to the Author)

Reviewer #5

(Remarks to the Author)

Reviewer #6

(Remarks to the Author)

Version 1:

Reviewer comments:

Reviewer #1

(Remarks to the Author)

In this revised version, the authors have adequately addressed all my comments and answered my questions. I remain very positive about this study which I find really exciting. I only have a few small additional suggestions and comments that would not require any further revision on my part.

Additional remarks:

In my opinion, the longer timescale used in figure R2 and S3B - with clear indication of the 'Lick-free' window - works better for showing DF/F signals and could be used in the main figures (including 3D and 7D-E).

I also find figure R3 very informative (it provides an easy and simple assessment of the performance of the decoder). It could be included in the supplementary figures if possible.

I am a bit confused by the explanation regarding the difference in baseline between the 'NoGo' and '>1 Hit' trials for the 'prior Hit different whisker' trials: "The '>1 prior Hit different' data came less from HiP blocks". Does that imply that the averaged traces presented in figure 3D come from different sessions/cells? I would assume that the best comparison between 'prior Hit same' and 'prior Hit different' trials would be for the same cells and same sessions? If that is not possible, shouldn't it nonetheless be possible to compute average responses for NoGo trials that match the condition, i.e. if different sessions are used for 'prior Hit same' and 'prior Hit different' trials, then use the corresponding NoGo trials (computed from the same sessions/cells)? I do believe that has no impact on the findings but should at least be clarified in the Methods section or figure legend.

Lines 247-248: "An exception was when attention was cued upward, which caused little upward CoM shift." In my opinion the most striking exception to the rule is the (r,-) condition which results in a prominent upward and caudal shift...

Minor:

line 129-130: "We also observed shifts in criterion (Δc) which with a more modest whisker-specific component" ?

Sylvain Crochet

Reviewer #2

(Remarks to the Author)

The authors have answered most of our questions, and taken away the main concerns. The manuscript has been improved and the expanded description of the behavioral measurements make it now easier to get an intuition for the effects. The point is well taken that d' provides a better control for the overall behavioral tendencies. Nonetheless, we remain to think that the probability of licking upon different priors is a more straightforward way to express the effect -- which is what the authors do anyway in figure 1E and G to support their claims. Altogether, we are supportive of this very interesting and complete paper, which also includes exciting leads for further research. We have no further comments.

Reviewer #3

(Remarks to the Author)

I have no further comments, and congratulate the authors on an excellent and further improved manuscript.

Reviewer #4

(Remarks to the Author)

Reviewer #5

(Remarks to the Author)

Reviewer #6

(Remarks to the Author)

February 18, 2025

Response to Reviewers

We thank the reviewers for their comments. To address them, we have added substantial new analysis and updated the text and figures. Our response to each comment is below, with the reviewer comments in blue, and our responses in black.

We provided responses to the editorial comments in the cover letter, as requested.

--Dan Feldman

Reviewer #1 (Remarks to the Author):

This is an excellent and exciting study from Daniel Feldman's lab. Over the past several years, Feldman's lab has been exploring different aspects of spatiotemporal coding in the mouse whisker system revealing complex interactions between patterns of multi-whisker sensory inputs and receptive fields in the barrel cortex. In this new study, the authors designed a clever behavioral paradigm and analyses to study the impact of trial/reward history on spatial attention in a whisker detection task. By randomly presenting the whisker stimulus to one of nine individual whiskers, the authors show that consecutive successful detections (Hit trials) of the whisker stimulus applied to the same whisker result in increased detectability if the next stimulus is applied again to that same whisker, but result in a slight decrease in detectability if the stimulus is applied to a different whisker. This shift in spatial attention to the last rewarded whisker cannot be explained by a difference in motor activity or general arousal, and it is correlated with an increased responsiveness of cortical neurons responding to the attended whisker. This response enhancement was more prominent in neurons in the surrounding barrel columns and resulted in a receptive field shift towards the attended whisker. The authors also investigate the possible contribution of VIP interneurons in mediating this trial-history-related change in neuronal response, finding a correlation between VIP neuron activity and general motor activity and arousal, but not with the whisker-specific shift in attention. Overall, the results are very compelling and exciting, and I have only a few questions and suggestions for clarification:

1) All along the study the authors use different versions of the task, using no delay, a 0.5 s or a 1.0 s delay, using different stimulus amplitudes (figure 1F and S1F). It is not always easy to understand which sessions (i.e. version of the task) are used for the different analyses. The authors should try to clarify this point whenever appropriate.

This is now clarified in Methods (quoted below) and in **Figure S1** legend. Attentional effects were similar in all task versions (**Fig. S1G-H**), which were combined for analysis. The number of mice trained on each task version is now stated in Methods (line 565). For delay periods, different delay periods were used in different subsets of mice (shown in **Fig. S1C-D**). Mice used in imaging or extracellular recording experiments had a delay period of 500 ms (for all spike recording mice and 7/14 2p imaging mice) or 1000 ms (for the other seven 2p imaging mice). This was used to temporally separate sensory-driven neural activity from later action- and reward-related activity. Three additional mice used only for behavioral data collection were tested without a delay period, to enable testing of attention effects on lick response latency (**Fig. S1E**). This is stated in Methods (lines 534-539) and **Fig. S1** legend.

The new Methods text (line 553-568) says:

Task variations

We trained mice on three variations of the task, schematized in **Fig. S1G**. All mice were initially trained on an "equal probability" (EqP) version of the task in which whisker identity on each Go trial was randomly selected from nine possible whiskers with equal 1/9 probability (EqP). The two other task variations were performed in a subset of sessions in some mice, and manipulated either the global or local probability of each specific whisker, while still randomly selecting whisker identity on each trial. In high probability (HiP) sessions, we manipulated the global stimulus probability of each whisker by presenting one whisker with higher probability (80% of Go trials) than the others. This was done in 300-400 trial blocks, interleaved with standard EqP blocks on the same day. In high probability of same whisker (HiPSame) sessions, we manipulated the local probability of repeating the same whisker stimulus on consecutive trials, while maintaining the overall probability of each whisker at 1/9. This was done in blocks interleaved with EqP blocks.

All mice were tested on the EqP version, and either the HiP/EqP block design (13 mice), or the HiPSame/EqP block design (7 mice). Whisker-specific attentional cueing was observed in all 3 task variants (**Fig. S1H**), so data from all variations were combined for the rest of the analyses.

2) In the same line, the use of different stimulus amplitude is briefly mentioned in the Results section but is not described in the Methods section. The authors should describe more clearly when stimuli of different amplitudes are used and the respective proportion of trials for each amplitude. The authors could also show a psychometric curve in the supplementary information.

The Methods (under 'Task Structure', quoted below) now explains that different whisker deflection amplitudes were interleaved in each session, in order to explore attentional modulation as a function of stimulus amplitude, and reports the proportion of trials for each stimulus amplitude. The psychometric curve for detection performance as a function of stimulus amplitude was shown in **Fig. S1F**. This panel shows that detection performance plateaued for amplitudes > 50-100 μm , and that trial history modulates detection performance at all amplitudes.

The Methods text now states (lines 539-546):

Different whisker deflection amplitudes were interleaved in each session, in order to explore attentional modulation as a function of stimulus amplitude (**Fig. S1F**). Across all behavioral experiments, the proportion of Go trials at each amplitude were: 0-100 μm : 5.2%, 100-200 μm : 25.5%, and 200-300 μm : 69.3%. Across all imaging experiments, the proportions were 0-100 μm : 0.5%, 100-200 μm : 15.9%, 200-300 μm : 83.7%. Because trial history modulated detection performance at all stimulus amplitudes (**Fig. S1F**), Go trials of all amplitudes were averaged to calculate mean behavioral and neural modulation as a function of trial history.

3) Lastly, if whisker stimuli of different amplitudes are always used, how is this accounted for in the analyses? Are all trials used, regardless of stimulus amplitude, to calculate d' or neuronal evoked responses? Are stimulus amplitudes accounted for in the trial history?

Since we empirically found that behavioral performance plateaued for amplitudes > 50 μm (**Fig. S1F**), we included all Go trials irrespective of amplitude when calculating d' and evoked neural responses. This is now stated in Methods (quoted below). Trial history-dependent modulation of d' was observed at all amplitudes (**Fig. S1F**).

The Methods now states (lines 544-546):

Because trial history modulated detection performance at all stimulus amplitudes (**Fig. S1F**), Go trials of all amplitudes were averaged to calculate mean behavioral and neural modulation as a function of trial history.

4) On another topic, the authors should show longer traces for evoked responses (e.g. Figure 3D). The baseline in particular is very short. For PYR L2/3 neurons, there seems to be a positive trend in Nogo trials on some traces. It might be interesting to consider a detrending approach similar to the one used for VIP neurons in Figure 7E?

As requested, we updated **Fig. 3D**, **Fig. 7D-E**, **Fig. S3**, and **Fig. S4** to show longer pre-stimulus baselines (1 sec instead of 0.5 sec as shown previously). These show very similar patterns as in the prior version (**Fig. R1**, left, reproduces the new **Fig. 3D**). As the reviewer points out, there is a positive trend in the baseline and throughout the trial for the >1 Hit history condition, which we discuss in the paper (e.g., lines 198-201), and which we interpret as an expectation or arousal effect that occurs when the mouse is rapidly accruing rewards. Even without detrending, the mean traces make clear that >1 prior Hits increases current-trial responses to the same whisker, but not to a different whisker (**Fig. 3D**). We debated, but chose not to detrend the PYR traces, in order to stay closest to the raw data. This whisker-specific attention effect is apparent even without detrending, and our quantification of evoked $\Delta F/F$ magnitude already subtracts the immediate prestimulus baseline $\Delta F/F$, which is similar to

Figure R1. Trial history effects on whisker-evoked $\Delta F/F$ for L2/3 pyramidal cells. Left, the new Fig. 3D, compiled from all block types. Right, HiP blocks have different pre-stimulus baseline $\Delta F/F$ dynamics, related to more vigorous reward licking in HiP blocks. detrending.

There is one condition in which the longer prestimulus baseline traces differ between current Go and NoGo trials (>1 prior Hit to a different whisker) (**Fig. R1**, left). Digging into the data, we found that this was because $\Delta F/F$ traces during the prestimulus period were different between our high-probability (HiP) block design and our other block designs, reflecting slightly more vigorous licking during the reward period in HiP blocks (**Fig. R1**, right). The blocks did not differ in other ways, and all showed the whisker-selective attention effect (**Fig. S1G-H**), and thus were combined for analysis (which maximizes statistical power). The '>1 prior Hit different' data came less from HiP blocks, which explains the difference in baseline. We consider this a minor discrepancy and still show the combined data across all block types in Fig. 3D.

We weren't sure if the reviewer was also asking for longer traces in the post-stimulus period. In the main figures, we only display and analyze the 0.7-sec lick free period, to avoid lick contamination of neural responses. But we now also show a longer 2-second post-stimulus period in the **new Fig. S3B** (reproduced here as **Fig. R2**). This includes both the lick-free 0.7-sec period and the next 1.3 sec, which may contain licks. This reveals the expected DF/F decline after the whisker stimulus, and shows all the same history-dependent features as the shorter, lick-free traces.

Figure R2. Trial history effects on whisker-evoked DF/F for L2/3 pyramidal cells, showing long post-stimulus period that includes both early lick-free, and late lick-contaminated periods. Plotting conventions as in main text Fig. 3D.

5) Regarding the quantification of movement in Figure 2A, I fail to understand the reason for subtracting the baseline activity 0.25 s before the onset of the whisker stimulus. It seems to me that it would be interesting to know if there is a difference in movement depending on trial history when the whisker stimulus is presented?

Thank you for this comment. We have now removed the prestimulus baseline subtraction of movement values from **Fig. 2A**. The results are the same: while any prior hit elevated whisker movement, body movement, and pupil size, movement traces were identical for prior same and prior different conditions, both for the last 3 seconds of the intertrial interval and for the stimulus period of the next trial. Thus, whisker-specific attentional modulation does not involve differences in body movement or arousal between prior same and prior different history conditions. This is presented in Results (lines 158-178).

While addressing the reviewer's question, we noticed we had accidentally included some behavioral sessions with poor signal-to-noise for DeepLabCut tracking of whisker position. We removed these sessions, which did not substantially affect the result. The new traces in **Fig. 2** and quantification in **S2** reflect this fixed analysis. The differences in prior trial lick rate in the >1 Hit condition reflect the same issue as in Point 4, in which 'prior >1 different' data come less from HiP blocks, and thus have less vigorous licking.

6) In Figure 4A, why not show the change in evoked response for consecutive hits for the CW whisker (central gray panel)? I believe it is shown in a separate figure in S5A?

Thank you for the suggestion. We have moved this supplemental panel to main **Fig. 4D**. This brings a new focus to **Fig. 4** on tuning changes during attention to the CW. To flesh this out, we also performed a new analysis asking whether attend-to-CW sharpens single-neuron tuning to the CW. We found that

it does, and have added new panels **Fig 4E-F** showing this. This means that during attention, we observe i) increases in sensory gain, ii) shifts in tuning toward the attended whisker, and iii) sharpening of tuning in the attended whisker's column -- all three major forms of tuning change observed during classical forms of attention in primates. A new paragraph in Results describes these findings (lines 259-266):

When the mouse attended to the CW, responses to that whisker within its S1 column increased, but there was no obvious shift in tuning peak (**Fig. 4D**). Instead, neurons increased their responsiveness to the CW, and decreased their responsiveness to the strongest SWs (**Fig. 4E**), thus increasing their preference (tuning sharpness) for the CW (**Fig. 4F**). Thus, attentional cueing involves receptive field shifts, tuning width changes, and modulation of whisker response magnitude, as observed during classical attention studies in primates^{18,30-32}. The mean tuning changes, combining all three of these effects, that occur in a S1 column as the mouse shifts attention to each nearby whisker are shown as receptive field contour plots in **Fig. S5C**.

7) For the decoder part of Figure 5, my understanding is that the detection of the whisker stimulus is based solely on the average activity of the population for a given trial. I wonder whether a decoder that better takes into account the complex dynamics of the population – that is, the activity of each individual neuron – might not achieve better performance. In particular, I am a little surprised by the decoder's high level of "false alarms" on CR trials (nearly 40%) and the low level of correct detection on Miss trials (less than 50%).

We agree, and have replaced our prior neural decoding analysis with a new decoder that individually weights each neuron in the population using logistic regression. To do this, we built a generalized linear model ('cvglmnet' in Matlab) with weights fit using ridge regression (L2-penalized logistic regression, $\alpha = 0$; optimal lambda estimated for each session using 10-fold cross-validation) to predict the presence or absence of a whisker stimulus on each trial from single-trial neural activity ($\Delta F/F$) of individual ROIs imaged in each session. A separate decoder was fit for each session. Decoder performance was tested on held-out trials using k-fold cross-validation with 10 randomly assigned folds. The results are now shown in the new **Figure 5**, and the decoder design is explained in Methods (lines 842-860). The new decoder shows higher overall performance ($68.6 \pm 0.9\%$ correct) and significant trial history effects. When the decoder was tested on different whiskers, we observed that prior Hits improved the ability to decode whiskers that were not the field best whisker (FBW, which is a proxy for the columnar whisker), but did not improve detection of the FBW. These are the same results as with the previous decoder, but now with a stronger population model. The Results presents the new decoder findings (lines 269-294).

The new decoder shows a FA rate of 27% for NoGo trials (similar to the mice), and a Hit rate of 60% for any whisker and 89% for the FBW (**Fig. 5B**). These are substantial improvements over the prior decoder. Imperfect performance of the decoder reflects the fact that neural population activity ($\Delta F/F$) on NoGo trials has some overlap with Go trials, especially Miss trials. As a result, when decoder performance is analyzed separately for behavioral Hit, Miss, FA and CR trials, stimulus detection (of any whisker) by the decoder is better for current Hit trials than current Miss trials (**Fig. R3**).

Figure R3. Decoder performance analyzed separately for behavioral Hit, Miss, False Alarm, and Correct Reject trials. Each point is the decoder for one imaging session. The decoder was trained on all trial types, but tested separately on the 4 trial types.

Sylvain Crochet

(Please do leave my name apparent to the authors)

Reviewer #2 (Remarks to the Author):

This very interesting paper investigates how prior reward history influences behavioral performance and neuronal responses in mouse S1. Using a Go/NoGo whisker deflection detection task, the authors demonstrate that recent associations of a reward with a whisker stimulus leads to the enhanced detection probability of the next whisker stimulus. The authors attribute this to an enhancement of spatially selective attention. They then use 2-photon imaging and spike recordings to show that whisker-evoked L2/3 and L5 pyramidal neural responses are also enhanced in the corresponding cortical columns if that whisker had been stimulated in prior trial and resulted in a reward. They then show that VIP interneurons, whose activity is often associated with arousal and attention, are not modulated by the reward history but instead respond to arousal and global behavioral state. This reward history-driven response modulation offers an interesting novel framework for studying behaviorally relevant sensory processing in cerebral cortex.

This is a very interesting study both from the perspective of reward history-associated perception, as well as for our understanding of sensory processing in cortex. I applaud the thoroughness of this study, the various angles of analyses as well as the depth at which the authors have gone through the data. At a first read, I found all aspects and findings intuitive and convincing. Upon a second and third read I found myself getting somewhat confused by a couple of points, which I am sure the authors will be able to clarify or correct.

1a. The main issue that has caused confusion is the use of d-prime (d') to indicate the instantaneous behavioral performance on a particular trial (which they term 'current' trial, e.g. from Fig. 1D onward). In my understanding, and as the authors describe in the methods, d' is derived from a series of 50 trials around the 'current trial'. This makes sense if one were interested in the general slowly evolving fluctuations in behavioral performance (i.e. signal sensitivity over a series of trials, such as in Fig. S1A). But I fail to see the relevance of this measure for estimating the effect of reward history (i.e. prior Go/hits). Since d' is based on both hit rates (HR) as well as false alarm rates (FR) over many trials, the measure is not a clean indicator of the impact of the nearest prior trials. The 'probability of lick on current trial' with various priors (Fig. 1E) is in my opinion the only measure that cleanly indicates the effect of reward history.

This is not how we calculated d-prime for the attention effect, so let us clarify our method, which we think is the source of the misunderstanding. We only used the sliding d-prime analysis (using a sliding bin of 50 trials) to identify poor-performance periods at the beginning and end of each session that reflect satiety effects, and to exclude them from further analysis. We then analyzed history effects within the entire remaining period, called the 'analysis period' (not in a sliding window). To calculate d-prime for a specific trial history, we 1) identified all Go trials with that trial history in the analysis period, and all NoGo trials with the same trial history in the analysis period, and then 2) calculated d-prime from the Hit rate and FA rate over this entire set of Go and NoGo trials. For example, to calculate d-prime for 'prior trial = NoGo', we calculate Hit rate for all Go trials that had ≥ 1 prior NoGo during the entire analysis period, and FA rate for all NoGo trials that had ≥ 1 prior NoGo during the entire analysis period. Likewise, to calculate d-prime for 'prior 1 Hit same', we calculate Hit rate for all Go trials that had 1 prior Hit to the same whisker, and FA rate for all NoGo trials with 1 prior Hit (to any whisker). Thus, we are able to measure how recent trial history influences the average d-prime (by averaging across all trials with that history, spread throughout the session). This is now stated in Results (lines 99-100) and in Methods (616-635):

Results (lines 99-100):

For each trial history category, we quantified detection sensitivity (d') and criterion (c) from the mouse's hit rate and false alarm rate for all current Go or NoGo trials with that trial history in the session.

Methods (lines 619-639):

For each session, we first used a sliding d-prime to identify poor-performance periods at the beginning and end of the session that reflect satiety effects, and to exclude them from further analysis. To do this, d' was computed across trials over the entire behavioral session. A sliding d' cutoff (calculated over a 50 trial sliding window) was applied to the start and end of the session, and analysis was restricted between the first and last trial that met the threshold, which was termed the 'analysis period'. A standard sliding d' cutoff of 0.5 was used for all behavioral and imaging analyses. We tested d' cutoffs 0.5, 0.7 and 1 to ensure that choice of d' cutoff did not affect key results. A sliding d' cutoff of 1.2 was used for extracellular recording analyses.

Next, we analyzed trial history effects within the analysis period of each session. To calculate d-prime for a specific trial history, we 1) identified all Go trials with that trial history in the analysis period, and all NoGo trials with the same trial history in the analysis period, and then 2) calculated d-prime from the Hit rate and FA rate over this entire set of Go and NoGo trials. For example, to calculate d-prime for 'prior trial = NoGo', we calculated Hit rate for all Go trials that had ≥ 1 prior NoGo during the entire analysis period, and FA rate for all NoGo trials that had ≥ 1 prior NoGo during the entire analysis period. Likewise, to calculate d-prime for 'prior 1 Hit same', we calculated Hit rate for all Go trials that had 1 prior Hit to the same whisker, and FA rate for all NoGo trials with 1 prior Hit (to any whisker). This method enables us to measure how recent trial history influences the average d-prime (by averaging across all trials with that history, spread throughout the session).

We use d-prime and criterion, rather than raw hit rate, because these separate the sensitivity of stimulus detection (d-prime) from the global behavioral response threshold (i.e., overall lickiness) (criterion) (for an overview, see Luo & Maunsell, 2019). We show that the elevation in d-prime from prior Hits is due an increase in Hit rate, not a decrease in FA rate (**Fig. 1g**). One major finding is that reward history selectively elevates d-prime for the previously rewarded whisker, not for other whiskers, which also indicates that it is not driven by changes in FA rate, since the same FA rate (from the same NoGo trials) are used in the d-prime calculation for prior Hit same and prior Hit different conditions (**Fig. 1d, 1h**).

1b. Further corroborating my point here, I find the values of d' and the deltas thereof very high (2.45 for the >1 Go/hit same whisker cases in Fig. 1D), which I think implies that if mice have HR on those trials of around 0.85 (roughly estimated as indicated in Fig. 1E), their FR should be around 0.08. For >1 Go/hit on different whisker cases, the d' is 0.82, but the HR on those trials isn't that much different, about 0.7 (Fig. 1E). That means that it is mostly the FR that has gone up in these sessions (to about 0.4). I think this point is more less reflected in Fig. 1G: for the 'prior different' data points, it is not so much the HR, but rather the FR that goes up. I feel that this is not very informative of the 'attention' effects of the prior Go/hits. Remarkably, the FR in all NoGo trials of Fig. 1E is never at 0.4, but mostly 0.2; so, I hope the authors can understand my confusion.

This occurs because d' is a nonlinear measure, so that d' calculated from the average Hit and FA rates is different than the average d' calculated from Hit and FA rate in each single session. (We have double-checked the calculations to confirm they are correct.).

Our analysis shows that the attention effect involves both an increase in d' for recently rewarded whiskers, plus a shift in criterion (increase in FA rate) that is not whisker specific, and instead reflects an increase in overall lickiness (on both NoGo trials and Go trials on all whiskers). **Fig. 1G** shows exactly this: For the 'prior same' condition, the dominant effect is an increase in Hit rate to the previously rewarded whisker, while for the 'prior different' condition, Hit rate stays flat or only modestly increases, while the increase in FA rate dominates. Note that we use exactly the same NoGo trials to calculate FA rate for 'prior same' and 'prior different' conditions, so the elevation of FA rate is identical for these conditions. The same patterns can be seen for the example sessions shown in **Fig. 1e**.

To reassure the reviewer that the attention effect can be directly observed in lick rate on Go trials, we replot the data in **Fig. 1g** (here by session, rather than by mouse) to show that attention to a previously rewarded whisker drives a whisker-specific increase in Hit rate to that whisker (**Fig. R4**).

Reviewer Fig. 4. Attention drives a large increase in Hit rate on Go trials to the recently rewarded whisker. Prior same = current trial is a Go trial on the same whisker as the last Hit trial. Prior different = current trial is a Go trial on a different whisker as the last Hit trial.

1c. The use of d' also includes the risk of intrinsic bias because the ' > 1 hits' cases may include more Go trials than the other cases, and particular Go/hit trial sequences are probably used repeatedly to calculate d' for different 'current' trials. What happens in a series of many consecutive Go trials when the mouse is at the top of its performance? In these cases, d' prime will rely only on very few NoGo trials, and hence the FR will be low but very capricious.

We hope that the improved description of our d-prime calculation method (under 1a, above) clears this up. Go and NoGo stimuli are presented randomly intermixed, not contingent on the mouse's performance. We are not analyzing d-prime in sliding windows, so we do not sample different numbers of Go or NoGo trials for the different history conditions. In addition, the principal effect that we focus on is the whisker-specific increase in d-prime following prior > 1 hit to the same whisker, but not a different whisker -- this cannot be an artifact of the number of NoGo trials, because we use exactly the same NoGo trials (prior >1 hit to any whisker) in the calculation of d-prime for 'prior same' and 'prior different'.

I am sure the authors have thought deeply about this and will be able to provide solid arguments for using the d' measure. It just was not intuitive to me, and I therefore encourage them to introduce and support the methodology more elaborately.

We hope the discussion above resolves the concerns. We have rewritten and expanded the Methods paragraphs describing the d-prime calculation (under 'Behavioral Analysis', lines 609-635, quoted here) , so that this is clearer for readers.

Behavioral Analysis

Behavioral performance was assessed using the signal detection theory measures⁹⁰ of detection sensitivity (d') and criterion (c), calculated from Hit rates (HR) and False Alarm rates (FA), as per their standard definitions:

$$d' = Z_{HR} - Z_{FA}$$
$$c = \frac{1}{2} (Z_{HR} + Z_{FA})$$

where Z is the inverse cumulative of the normal distribution.

For each session, we first used a sliding d-prime to identify poor-performance periods at the beginning and end of the session that reflect satiety effects, and to exclude them from further analysis. To do this, d' was computed across trials over the entire behavioral session. A sliding d' cutoff (calculated over a 50 trial sliding window) was applied to the start and end of the session, and analysis was restricted between the first and last trial that met the threshold, which was termed the 'analysis period'. A standard sliding d' cutoff of 0.5 was used for all behavioral and imaging analyses. We tested d' cutoffs 0.5, 0.7 and 1 to ensure that choice of d' cutoff did not affect key results. A sliding d' cutoff of 1.2 was used for extracellular recording analyses.

Next, we analyzed trial history effects within the analysis period of each session. To calculate d-prime for a specific trial history, we 1) identified all Go trials with that trial history in the analysis period, and all NoGo trials with the same trial history in the analysis period, and then 2) calculated d-prime from the Hit rate and FA rate over this entire set of Go and NoGo trials. For example, to calculate d-prime for 'prior trial = NoGo', we calculated Hit rate for all Go trials that had ≥ 1 prior NoGo during the entire analysis period, and FA rate for all NoGo trials that had ≥ 1 prior NoGo during the entire analysis period. Likewise, to calculate d-prime for 'prior 1 Hit same', we calculated Hit rate for all Go trials that had 1 prior Hit to the same whisker, and FA rate for all NoGo trials with 1 prior Hit (to any whisker). This method enables us to measure how recent trial history influences the average d-prime (by averaging across all trials with that history, spread throughout the session).

2. Related to point 1, the authors mention in the methods that 'Mice were deemed task experts when they exhibited stable performance at $d' > 1$ '. However, the d' in expert mice was on average 0.923 which corresponds to an approximate fraction of correct trials of 0.6 -slightly above the chance level.

Does this mean that the data presented in this study also pertain to mice that are still in the process of learning the association between any whisker stimulation and the reward. In the literature, mice are considered experts in Go/No-Go tasks when d' exceeds at least 1.5. Could the authors comment on that?

We trained all mice to an average $d' > 1$, assessed across several days. This is a challenging task, because stimuli are applied on random whiskers, rather than consistently on the same whisker each trial (as is the case for most passive whisker detection tasks). In addition, most mice were trained with a 500-1000 ms delay period, which is well known to decrease performance for mice on many tasks. Behavioral performance plateaued for most mice after reaching $d'=1$, and rather than overtrain mice, we began data collection. Daily performance did fluctuate, which is why the mean d' was 0.923 across all sessions for all mice reported here.

3. In figure 2A-B, the whisker motions for NoGo trials seems also sensitive to prior trial history. Is this associated with a higher FR? In figure 2B, the magnitude of the change of whisker position change is similar in all 3 trial types, including for NoGo trials. Can this be explained by an overall increase in compulsive licking behavior when mice experience several consecutive Hits on the same whisker?

The reviewer is correct that prior trial history affects whisker motion in the inter-trial interval before the next trial (these are the time points at $time < 0$ in **Fig. 2a**, and quantified in **Suppl. Fig. 2b**). This likely reflects arousal following reward retrieval, which takes seconds to dissipate, because the same effect is seen for body motion and pupil size (see lower panels of **Fig. 2a**). These effects in the inter-trial interval must occur for all 3 trial types, because the mouse does not yet know whether the next trial is a Go (same whisker), Go (different whisker), or NoGo trial. There is a similar non-specific trial history effect on stimulus-evoked whisking and body motion after the whisker stimulus is delivered at the start of the next trial (time points > 0 in **Fig. 2a**, quantified in **Fig. 2b**).

This is explained in Results (lines 160-169):

Reward retrieval at the end of Hit trials was associated with whisker movement, body movement (detected from platform motion), and pupil dilation that slowly subsided during the ITI before the next trial (**Fig. 2A**). The magnitude of movement and pupil dilation during the ITI was greater after 1 or >1 prior Hits, relative to prior NoGo, but was identical for prior same and prior different conditions (**Fig. 2A, Fig. S2B**). Thus, mice exhibited increased motion and arousal following prior Hits. During the subsequent trial, whisker stimulation evoked modest whisker and body motion during the stimulus period, and these were also heightened after prior Hits ($p = 1e-4$, permutation test), indicating that behavioral arousal and motion effects from prior Hits persisted into subsequent trials, but did not differ between prior same whisker and prior different conditions ($p = 0.34$ and 0.76 , permutation test **Fig. 2A-B**).

These trial history effects on whisker motion are not due to compulsive licking that extends into the next trial, because we require a 3-sec no-lick period for initiation of each trial. Thus, our interpretation is that licking to retrieve reward, and the arousal that follows for many seconds, increases whisker motion in the ITI and stimulus-evoked movement on subsequent trials, but that compulsive licking through the ITI is not the cause. The underlying cause is likely to be arousal state, which may correspond to the criterion shift we observe with prior rewards and the increased VIP cell activity we see in **Fig. 7**. It cannot contribute to the whisker-specific effect on detection sensitivity (d' -prime).

4. Related to my previous point, the activity of L2/3 PYR neurons is higher in NoGo trials when it was preceded by multiple Hits (Figure 3D and E). As argued under point 3, could this mean that most of the NoGo trials in the 'Prior trial >1 Hit' condition are FAs whereas most of NoGo trials in the 'Prior trial

NoGo' condition are CRs? If one were to consider that the repetition of Go/hits on the same whisker could provoke a transiently more compulsive licking behavior, the interpretation of the data in this study would be very different. Can the authors exclude this? I would like to invite to comment on this in the paper as well.

There are two effects observed in the L2/3 PYR responses with trial history. The first is that on Go trials, whisker-evoked responses are elevated following 1 or > 1 prior Hit to the same whisker, but this doesn't occur for prior Hits to a different whisker. This whisker-specific effect on neural responses matches the whisker-specific effect on d-prime. The second effect, as pointed out by the reviewer, is that on NoGo trials, L2/3 PYR cells show more activity with increasing number of prior Hits, which matches the criterion effect (i.e., correlates with higher FA rate). Our behavioral analysis shows that attention involves both a d-prime effect and a criterion effect, consistent with prior attention literature in primates (e.g., Luo & Maunsell, 2015, 2018, 2019). If the only behavioral effect were the criterion effect, then the trial history effect on neural responses in NoGo trials could simply reflect higher FA rate. But our focus is the whisker-selective d-prime effect and its correlate in S1. We now explicitly discuss the difference between the whisker-specific effect on PYR sensory gain (which is somatotopically precise in S1) and the non-whisker specific increase in PYR baseline activity which may mediate the criterion effect (Discussion, lines 404-413):

This somatotopically restricted boosting⁶¹⁻⁶⁷ is distinct from the spatially broad modulation of sensory responses that occurs across entire cortical areas (or multiple areas) in response to global behavioral state (e.g., arousal indexed by pupil size, active whisker movement for S1, or locomotion for V1)⁴⁵. Our results show that attentional boosting can be flexibly targeted with a precision of ~300 μm in cortical space for stimulus-specific modulation of the neural code. Thus, neural control circuits for attention (which may involve feedforward, local, feedback, or neuromodulatory circuits) must operate with this spatial precision. In addition to this somatotopically precise boost in PYR cell sensory gain, recent rewards also drove a generalized increase in activity on subsequent NoGo trials, which was not whisker-specific and may contribute to the non-whisker specific shift in behavioral criterion.

5. On line 134: "Trial history effects were driven by stimulus-reward association, not by stimulus salience or reward probability, because whisker deflections were physically identical, and reward probability was always 100% for each whisker." Shouldn't the authors report the probability of licking for all 9 whiskers to make this statement? Even though the whiskers' deflections are identical, the detection and perception of stimuli on different whiskers might substantially differ.

This statement was only meant to convey that the physical deflection of each whisker was identical. The attentional effect was observed, with very similar magnitude, for all 9 whiskers that were tested (**Fig. 1M**). (That is, Hits to each of these whiskers generate a similar d-prime effect on subsequent trials.) So any subtle differences in perceptual salience between those 9 whiskers did not meaningfully affect the attentional effect.

In line 259, the authors state that they "observed modest, above-chance performance for detecting any of the 9 whiskers". This claim is made without statistically testing the performance against the chance level of 50%.

This refers to the decoding figure (**Fig. 5**). We have replaced the previous decoder with an improved decoder that weights neurons individually, as described for Reviewer 1 Point 7. To assess whether decoder performance was above chance for detecting any of the 9 whiskers, we compared the 'any whisker' decoder to a decoder trained on shuffled data. This was significant ($p = 1e-4$), as shown in **Fig. 5b**.

6. In figure 5, how do the authors explain that the fraction of trials predicted as stimulus is around 40% for the NoGo condition? I find this value particularly high given that no stimulus is delivered.

The new decoder shows a FA rate of 27% for NoGo trials (similar to the mice), and a Hit rate of 60% for any whisker and 89% for the FBW (**Fig. 5B**). These are substantial improvements over the prior decoder. We don't show it in the paper, but decoder error rates are partially attributable to differences in cortical activity on Hit, Miss, FA and CR trials (please see **Fig. R3** under Reviewer 1, Point 7).

7. How do the authors explain, in Fig. 5E-G, that the fraction of trials detected as stimulus is not improved when there was 1 Hit on the same whisker previously?

The new decoder shows that same behavior -- stimulus detection on single trials is strongly boosted following >1 Hit to the same whisker, but not following 1 Hit to the same whisker (**Fig. 5C, E**). We interpret this as indicating that the attentional boost in neural activity after 1 Hit is strong enough to be observed in mean PYR firing rates, but is still modest relative to single-trial variability, so that decoder performance is not improved. This is now discussed in Results (quoted below). This interpretation is generally consistent with our other analyses which report a relatively modest increase in whisker-evoked DF/F after 1 prior same Hit, but a much larger increase after >1 prior same Hit.

The Results now states (lines 292-293):

We interpret the lack of improvement in whisker decoding after a single Hit (**Fig. 5C-E**) to mean that 1 prior Hit boosts whisker responses only modestly relative to single-trial variability within the imaging field.

8. In figure 7D, the activity traces of VIP neurons are somewhat puzzling. The authors state that the declining baseline is the result of prior trial activity. However, in these examples, the transient evoked by the whisker stimulation is relatively small and the average activity keeps declining, below baseline levels, about 1.5 s after the whisker stimulation. This must mean that there is a second large peak of activity occurring during the reward period. Moreover, I don't understand how multiple NoGo trials can produce a constantly declining baseline (dashed line of left examples). Can the authors comment on this, and perhaps they could show example traces that extend into the reward period and the next trial?

The reviewer is correct that the declining baseline in VIP cells is due to the fact that VIP cell activity tracks multiple aspects of movement and arousal, which peak at the end of each prior trial and decline to the start of the current trial. We analyzed this in detail in a prior paper (Ramamurthy et al., 2023). We have now added a new **Figure 7C** that shows this behavior (reproduced here as **Reviewer Fig. 5**). We showed previously that in S1, VIP cells are activated by whisker stimuli, arousal (indexed by pupil size), whisker and body movements, and very strongly by goal-directed licking (Ramamurthy et al. 2023). When the prior trial was a NoGo or a Miss, there can be a certain amount of fidgeting and inter-trial licking, which subsides before the current trial start because we require a 3-sec no-lick period to start each trial. As a result, VIP cell activity is declining at the beginning of each trial (**Fig. 7C**). When the prior trial was a Hit, there is vigorous licking that strongly activates VIP cells, causing VIP activity to decline even more strongly throughout the ITI and at the beginning of the next trial (**Fig. 7C**). This explains the decreasing VIP baseline and its trial history dependence. We now explain this more clearly in the paper (Results and new **Fig. 7C**).

Figure R5. Behavioral dynamics in the inter-trial interval and its relationship to VIP DF/F. The top 4 panels show the dynamics of different body movements in the last 3 seconds of the inter-trial interval (ITI) prior to the start of the current trial. Mean dynamics are calculated separately for prior trial = Hit and prior trial = NoGo or Miss conditions. Whisker motion, body motion and Δ pupil size are from DeepLabCut analysis of behavioral videos. Licks are lick counts. The bottom panel shows mean VIP DF/F in S1 for the two history conditions. VIP Δ F/F declines in the ITI leading up to each current trial, and this decline is strongest after prior Hit trials, because these trials end in intense licking accompanied by body motion and arousal, which then decline over time prior to current trial start. This figure is the new **Fig. 7C**.

The Results explains this interpretation (lines 336-349):

VIP cells in S1 are activated by arousal (indexed by pupil size), whisker and body movement, and goal-directed licking during this whisker detection task⁴⁴. These behaviors all peak at the end of Hit trials, as mice retrieve rewards, and then systematically decline during the ITI, which ends with a 3-sec lick-free period that is required to initiate the next trial. As a result, VIP cell Δ F/F falls systematically during the ITI after Hit trials, correlated with these behavioral variables, and falls less after NoGo or Miss trials (**Fig. 7C**). This creates a declining baseline for Δ F/F of VIP cells at the beginning of each current trial. VIP cells in S1 also show robust whisker stimulus-evoked Δ F/F transients, which ride on this declining baseline⁴⁴. To test whether whisker-evoked VIP responses are greater when reward history cues attentional capture, we calculated mean Δ F/F traces ($n = 7$ mice, 103 sessions, 1843 VIP cells, 7-31 sessions per mouse, mean 23.8 ± 0.61 [range 9-56] cells per mouse) as a function of trial history. Baseline (prestimulus) Δ F/F declined more steeply on trials following prior Hits than following prior NoGo or Miss, as expected. Superimposed on this, and clearest after detrending the baseline, whisker-evoked Δ F/F was also increased after prior Hit trials. However, this was not whisker-specific (**Fig. 7D-F**).

9. I think the authors make a convincing case for attention effects, but I am curious to hear if they considered short-term plasticity as a modulating process. If consecutive whisker stimuli plus a reward leave a short-lived (e.g. around 10 seconds) plasticity trace in the corresponding barrel column, one would expect a similar phenomenon, i.e. >1 Go/hit increases the pyramidal cell responses, which subsequently increases the detection probability. In this case the VIP neuron activity during the trial wouldn't be the modulating force but rather their activity in between the trials (i.e. at the rewards) – equivalent to an eligibility trace. If interesting, they could include this as a discussion point.

We have definitely considered this. Many cellular and circuit mechanisms have been proposed for various forms of attention, and this system will be powerful platform for investigating these mechanisms. We agree that short-lived plasticity could explain the behavioral and neural signatures of this form of attention, and some prior reviews have suggested that attention could involve short-term plasticity (e.g., Jääskeläinen et al., 2011). We don't discuss this in the paper, because it is too speculative and just one of many possible mechanisms at this point (including top-down projections, disinhibitory circuits, neuromodulatory circuits, and more), but we look forward to investigating this.

Minor points:

10. Figure 1D and other panels:

The design of the behavioral task is well explained, but the boundaries of the number of Go and NoGo sequences is not so clear (related to main point 1). Did the authors restrict the number of consecutive Go or NoGo trials?

We only restricted the number of consecutive Go and NoGo trials during one phase of training, when we introduced the delay period, to ensure the mice didn't lose motivation or alter their response strategy during this time. After the delay period was learned, we reinstated the pure random selection of Go and NoGo trials, with no limits on consecutive numbers, until mice reached behavioral criterion, and during data collection. Thus, for all of the data reported here, Go and NoGo trials were chosen purely randomly. This is now clarified in Methods (lines 575-579):

In all these task variations, the presentation of Go or NoGo trials, and the identity of the whisker on each Go trial, were chosen randomly for each trial, and was not contingent on the mouse's performance. The only exception was during a transient phase of training (Stage 6), when we briefly capped the maximum number of consecutive Go or NoGo trials, to help ensure mice did not alter their response strategy as they were learning the delay period.

11. For the reader to get a good intuition for the task, it might be helpful to move figure S1A to figure 1. This is a good suggestion, but we just can't figure out how to make this fit in the busy **Fig. 1**. We think it is appropriate in **Fig. S1** since it illustrates the fluctuations that occur in behavioral performance throughout a testing session in an expert mouse.

12. Figure S1B: Above the D2 whisker trial following C2 (Miss), should the color perhaps be red instead of blue

This has been corrected, thank you.

Reviewer #3 (Remarks to the Author):

Ramamurthy et al present a novel somatosensory spatial attention task and explore its cellular mechanistic basis. By exploiting the history of which spatially localised stimuli were rewarded they show that mice can be encouraged to shift their attention between specific whiskers across strings of trials. Through a combination of behaviour, 2-photon calcium imaging and electrophysiological recordings, the authors present compelling and well controlled evidence for both behavioural and neural correlates of attention during their task. The authors do well to emphasize the distinction between the more commonly studied bottom-up or top-down attention and this phenomenon of attentional capture guided by reward history.

Additionally, they examine the activity of VIP interneurons, which have been hypothesised to be key in attentional control. The authors find that VIP interneurons do not carry an attention signal, corroborating recent work in the visual system.

Overall this is a very exciting and important study which both establishes a novel attentional paradigm and also rules out a key cellular hypothesis.

We have a few points which if addressed by the authors would make the study even stronger.

1. It would be interesting to know what effect the reward cued focal attention in this task has on inter-neuronal correlations and how it compares to the effects observed in previous studies of attention.

We agree this is an interesting question. It is one of several that would be best studied with our spike recording data, which has much higher temporal resolution than the calcium imaging data.

Unfortunately, we only generated limited spike recording datasets for this paper, and we don't have

sufficient data for a thorough analysis of neuronal firing correlations. We plan on addressing this in a future study with a larger spiking dataset.

2. Is it possible that the spread in behavioural and neural effect of repeat stimulation onto the adjacent row and arc could in part be a result of the piezo array? That an active piezo subtly activates its neighbours?

This is a reasonable concern, but we know it's not generating the somatotopic spread of the attention effect. Mechanical coupling from one piezo to a within-row or within-arc neighbor in our array is only ~1% (measured by laser interferometry while deflecting the adjacent whisker). For a 250 μm deflection, this corresponds to ~2.5 μm peak deflection of the neighboring whisker, which we know is not sufficient to drive sensory afferent spikes (**Figure R6**). In addition, our attention effects are apparent not on the same trial as the original Hit, but on the subsequent trial. Thus, the behavioral and neural spread of the attention affect cannot be attributed to unwanted mechanical coupling in the piezo array.

Figure R6. Lack of mechanical coupling between whiskers in the piezo array. A, Piezo movement traces from a directly stimulated piezo in our 9-piezo array (using a ramp-return deflection with 240 μm amplitude) and from an adjacent piezo, measured by laser interferometry. B, Quantification of mechanical coupling between whiskers, quantified for peak amplitude of deflection. Coupling is only ~1% for ramp-return stimuli, indicating that piezos are mechanically well isolated.

3. There could be more emphasis in the text on the distinction between this phenomenon of attentional capture guided by reward history and bottom-up attention, particularly in the introduction section. Some discussion about how the neural boosting the authors have discovered compares to the existing literature on this type of attention.

Thank you for this suggestion. We have added a sentence to the introduction:

This effect is not driven by the physical salience of stimuli (classically termed bottom-up attention) but by their recent association with reward. [Introduction, para 1, lines 50-51]

To our knowledge, this is the first time that attentional boosting for value-guided attention has been observed at the single-cell level in cortex. Our results align with a prior fMRI study in humans that showed boosting of sensory-evoked activation driven by value (prior reward) on a trial-by-trial basis (Serences, 2008). We now cite this in the Discussion (line 388-390):

On the macroscopic scale, human brain imaging and focal pharmacological inactivation studies in non-human primates indicate that spatial attention in vision is retinotopically organized within visual cortical areas^{1,18,61-64}, including in a study of value-guided attention that showed enhanced sensory-evoked activation driven by recent prior reward in human V1⁶⁵.

Other minor points:

1. At some points, multiple nested samples are presented. For example, line 325 "(n = 7 mice, 103 sessions, 1843 VIP cells)" It would be helpful to know the range in sessions per mice and in VIP cells per session.

These ranges are now reported in Results (lines 343-346):

To test whether whisker-evoked VIP responses are greater when reward history cues attentional capture, we calculated mean $\Delta F/F$ traces (n = 7 mice, 103 sessions, 1843 VIP cells, 7-31 sessions per mouse, mean 23.8 ± 0.61 [range 9-56] cells per mouse) as a function of trial history.

2. The wording when discussing the decoding of neural activity could be clearer, specifically about what is being decoded. E.g. Lines 832-833 – is the decoder classifying which of the 9 whiskers was stimulated on a given trial? Or is it providing a binary answer of whisker stimulated vs no stimulation?

We have edited this section to make the wording clearer. The decoder is not predicting whisker identity, but only the binary of whether any whisker was stimulated in a given trial. We now state this explicitly in the Methods (lines 842-844):

We built a neural decoder that predicts the presence or absence of a whisker stimulus (i.e., whisker detection) from single-trial neural activity of all individual whisker-responsive L2/3 PYR cells in each 2p imaging session.

3. In supplementary figures S3E and S7 it's unclear what the dividing line in the columns is. The onset of whisker stimulation?

Yes, it represents the onset of whisker stimulation. The figures have been updated to label this.

4. Why is an n of 1843 VIP cells stated in the text and Figure 7E, but only 852 cells in Figure 7G and Figure S7?

The full dataset consists of 1843 VIP cells that were imaged in sessions that met all our stated criteria for data inclusion. This full dataset was used to generate population activity traces shown in **Fig. 7D-E**. Within this dataset, cells varied in having a large number of trials, or just a few trials, for a given history condition. For single-cell analysis, we identified a subset of these cells in which every history condition was sampled sufficiently to calculate a meaningful single-cell $\Delta F/F$ trace for each history condition, and to calculate attention metrics on the single-cell level (AMI). This was 852 cells, which were used to for the AMI analyses (**Figure 7G-H**) and to show $\Delta F/F$ traces for each history condition (**Figure S7**). This is now explained in Methods (lines 831-839):

N's for VIP cell analysis. The full VIP imaging dataset consisted of 1843 VIP cells that were imaged in sessions that met all stated criteria for data inclusion. This full dataset was used to generate population activity traces shown in Fig. 7D-E. Within this dataset, cells varied in having a large number of trials, or just a few trials, for a given history condition. For single-cell analysis, we identified a subset of these cells in which every history condition was sampled sufficiently to calculate a meaningful single-cell $\Delta F/F$ trace for each history condition, and to calculate attention metrics on the single-cell level (AMI). This was 852 cells, which were used to for the AMI analyses (**Fig. 7G-H**) and to show $\Delta F/F$ traces for each history condition (Figure S7).

Reviewer #4 (Remarks to the Author):

Reviewer #5 (Remarks to the Author):

Reviewer #6 (Remarks to the Author):

April 28, 2025

Response to Reviewers

We thank the reviewers for their comments. We have addressed them as shown below, by adding a supplemental figure, a new section in Methods, and by editing two sections in Results.

We believe this addresses all the issues.

Sincerely,

Deepa Ramamurthy & Dan Feldman

Reviewer #1 (Remarks to the Author):

In this revised version, the authors have adequately addressed all my comments and answered my questions. I remain very positive about this study which I find really exciting. I only have a few small additional suggestions and comments that would not require any further revision on my part.

Additional remarks:

In my opinion, the longer timescale used in figure R2 and S3B - with clear indication of the 'Lick-free' window - works better for showing DF/F signals and could be used in the main figures (including 3D and 7D-E).

We appreciate this view, but we also think there is value in having the figures consistently show only lick-free periods, which matches the analysis and the study design. We do show the longer timescale DF/F traces with lick-free and non-lick-free periods marked, in **Figure S3B**, so readers can see this as well.

I also find figure R3 very informative (it provides an easy and simple assessment of the performance of the decoder). It could be included in the supplementary figures if possible.

Thank you for the comment. We have now inserted Fig. R3 as a supplementary figure (new **Figure S6**). As a result, we have renumbered old Figures S6-7 as Figures S7-8.

I am a bit confused by the explanation regarding the difference in baseline between the 'NoGo' and '>1 Hit' trials for the 'prior Hit different whisker' trials: "The '>1 prior Hit different' data came less from HiP blocks". Does that imply that the averaged traces presented in figure 3D come from different sessions/cells? I would assume that the best comparison between 'prior Hit same' and 'prior Hit different' trials would be for the same cells and same sessions? If that is not possible, shouldn't it nonetheless be possible to compute average responses for NoGo trials that match the condition, i.e. if different sessions are used for 'prior Hit same' and 'prior Hit different' trials, then use the corresponding NoGo trials (computed from the same sessions/cells)? I do believe that has no impact on the findings but should at least be clarified in the Methods section or figure legend.

We apologize for not being clear. The full PYR imaging dataset consisted of 6906 PYR cells imaged in sessions that met our stated criteria for data inclusion. This full dataset was used to generate population DF/F traces in **Fig. 3D**. Within this dataset, cells varied in having a large number of trials for some history conditions, but could have too few trials for meaningful analysis in other history conditions (and thus were omitted from population analysis of those conditions). To ensure that this unequal sampling did not create biases in our results, we identified a subset of 2115 cells that had sufficient sampling to calculate a meaningful single-cell $\Delta F/F$ trace in every history condition. These are the cells

we used for AMI calculation at the single-cell level (**Fig. 3H-I**), and to show single-cell $\Delta F/F$ traces for each history condition (**Fig. S3G**). We recalculated the population average $\Delta F/F$ traces for these 2115 cells, where each cell is represented in each history condition. We observed virtually identical average $\Delta F/F$ traces as for the full population of 6906 cells in Fig. 3D. We now explain these methodological issues, and state that identical results were observed for the subset of cells sampled in all history conditions, in the Methods section, under 'N's for PYR and VIP cell analysis.' This revised section states:

N's for PYR and VIP cell analysis

The full PYR imaging dataset consisted of 6906 PYR cells and the full VIP imaging dataset consisted of 1843 VIP cells, that were imaged in sessions that met all stated criteria for data inclusion. These full datasets were used to generate the population activity traces in **Fig. 3D** and **Fig. 7D-E**, respectively. Within each of these datasets, cells varied in having a large number of trials for some history conditions, but could have too few trials for meaningful analysis in other history conditions (in which case they were omitted from population analysis of those conditions). For single-cell analysis, we identified a subset of these cells in which every history condition was sampled sufficiently to calculate a meaningful single-cell $\Delta F/F$ trace for each history condition, and to calculate attention metrics (AMI) on the single-cell level. This was 2115 PYR cells and 852 VIP cells, which were used to for the AMI analyses (**Fig. 3H-I**, **Fig. 7G-H**) and to show $\Delta F/F$ traces for each history condition for individual cells (**Fig. S3G**, **Figure S7**). We verified that attentional effects on population average $\Delta F/F$ traces shown with the full dataset (**Fig. 3D** and **7D-E**) were also observed, virtually identically, for this subset of cells that were sampled in all history conditions (data not shown).

Lines 247-248: “An exception was when attention was cued upward, which caused little upward CoM shift.” In my opinion the most striking exception to the rule is the (r,-) condition which results in a prominent upward and caudal shift...

We agree that (r,-) is also an exception. Overall, the smallest receptive field shifts occur when attention is cued within the CW row [(r,-) and (c,-)], and in the upward direction. We have updated the Results to reflect this (lines 249-253):

History-based cueing to an attentional target whisker generally shifted tuning CoM towards that whisker, as evident from comparing rostral vs. caudal or upward vs. downward attentional shifts (**Fig. 4B**). The magnitude of CoM shifts was smaller when attention was cued upward or within the CW row [(r,-) and (c,-) positions]. These differences was not explained by known experimental factors, but could reflect a spatial asymmetries in attentional effects³³ on whisker touch (**Fig. 4B**).

Minor:

line 129-130: “We also observed shifts in criterion (Δc) which with a more modest whisker-specific component” ?

While the main factor driving shifts in criterion is shifts in False Alarm rate, Hit Rate is a factor in the calculation of c , so Δc for the >1Hit Same history condition can be different than Δc for >1HitDifferent. We have edited the Results to explain this more clearly (lines 129-134):

We also observed overall shifts towards more liberal criterion values^{20,22-23} (Δc) due to the increased False Alarm rate (**Fig. 1I**). Δc had a modest whisker-specific component ($C_{>1HitSame}$ vs $C_{>1HitDiff}$, $p = 0.002$, permutation test) due to the influence of Hit rate on criterion calculation. On average, the whisker-specific shift in sensitivity was larger than the whisker-specific shift in criterion (**Fig. 1J**; $\Delta d'_{>1HitSame} = 1.4 \pm 0.20$, $\Delta d'_{>1HitDiff} = -0.47 \pm 0.11$, $p = 1.0e-4$, permutation test; $\Delta C_{>1HitSame} = -2.03 \pm 0.18$, $\Delta C_{>1HitDiff} = -1.34 \pm 0.08$, $p = 7.0e-4$, permutation test).

Sylvain Crochet

Reviewer #2 (Remarks to the Author):

The authors have answered most of our questions, and taken away the main concerns. The manuscript has been improved and the expanded description of the behavioral measurements make it now easier to get an intuition for the effects. The point is well taken that d' provides a better control for the overall behavioral tendencies. Nonetheless, we remain to think that the probability of licking upon different priors is a more straightforward way to express the effect -- which is what the authors do anyway in figure 1E and G to support their claims. Altogether, we are supportive of this very interesting and complete paper, which also includes exciting leads for further research. We have no further comments.

Reviewer #3 (Remarks to the Author):

I have no further comments, and congratulate the authors on an excellent and further improved manuscript.

Reviewer #4 (Remarks to the Author):

Reviewer #5 (Remarks to the Author):

Reviewer #6 (Remarks to the Author):
